

# Ensemble Variational Assimilation as a Probabilistic Estimator. Part I: The linear and weak non-linear case

Mohamed Jardak[1,2] and Olivier Talagrand[1]

[1]LMD/IPSL, CNRS, ENS, PSL Research University, 75231, Paris, France
[2]Data Assimilation and Ensembles Research & Development Group, Met Office, Exeter, Devon, UK

*Correspondence to:* M. Jardak (mohamed.jardak@metoffice.gov.uk)

**Abstract.**

Data assimilation is considered as a problem in Bayesian estimation, viz. determine the probability distribution for the state of the observed system, conditioned by the available data. In the linear and additive Gaussian case, a Monte-Carlo sample of the Bayesian probability distribution (which is Gaussian and known explicitly) can be obtained by a simple procedure : perturb the data according to the probability distribution of their own errors, and perform an assimilation on the perturbed data. The performance of that approach, called *Ensemble Variational Assimilation* (*EnsVAR*), is studied in the two parts of the paper on the non-linear low-dimensional Lorenz-96 chaotic system, the assimilation being performed by the standard variational procedure. In Part I, EnsVAR is implemented first, for reference, in a linear and Gaussian case, and then in a weakly non-linear case (assimilation over 5 days of the system). The performances of the algorithm, considered as a statistical estimator, are very similar in the two cases. Additional comparison shows that the performance of EnsVAR is better, both in the assimilation and forecast phases, than that of standard algorithms for Ensemble Kalman Filter and Particle Filter (although at a higher cost). Globally similar results are obtained with the Kuramoto-Sivashinsky equation.

## 1 Introduction

The purpose of assimilation of observations is to reconstruct as accurately as possible the state of the system under observation, using all the relevant available information. In geophysical fluid applications,



such as meteorology or oceanography, that relevant information essentially consists of the observations proper, and of the physical laws which govern the evolution of the atmosphere or the ocean. Those

physical laws are in practice available in the form of a discretized numerical model. Assimilation is therefore the process by which the observations are combined together with a numerical model of the dynamics of the observed system in order to obtain an accurate description of the state of that system.

All the available information, observations as well as numerical model, is affected (and, as far as we can tell, will always be affected) with some uncertainty, and one may wish to quantify the resulting uncertainty

on the output of the assimilation process. If one chooses to quantify uncertainty in the form of probability distributions (see, *e.g.*, Jaynes (2004), or Tarantola (2005), for a discussion of the problems which underlie that choice), assimilation can be stated as a problem in Bayesian estimation. Namely, determine the probability distribution for the state of the observed system, conditioned by the available information. That statement makes sense only under the condition that the available information is described from the

start in the form of probability distributions. We will not discuss here the difficult problems associated with that condition (see Tarantola (2005) for such a discussion), and will assume below that it is verified.

There is one situation in which the Bayesian probability distribution is readily obtained in analytical form. That is when the link between the available information on the one hand, and the unknown system state on the other, is linear, and affected by additive Gaussian error. The Bayesian probability distribution

is then Gaussian, with explicitly known expectation and covariance matrix (see Section 2 below).

Now, the very large dimension of the numerical models used in meteorology and oceanography (that dimension can lie in the range $10^6$ to $10^9$) forbids explicit description of probability distributions in the corresponding state spaces. A widely used practical solution is to describe the uncertainty in the form

of an ensemble of points in state space, the dispersion of the ensemble being meant to span the uncertainty. Two main classes of algorithms for ensemble assimilation exist at present. *Ensemble Kalman Filter* (*EnKF*), originally introduced by of Evensen (1994) and further studied by many authors (Evensen (2003) and Houtekamer and Mitchell (1998, 2001)), is a heuristic extension to large dimensions of the standard Kalman Filter (KF) (Kalman (1960)). The latter exactly achieves Bayesian estimation in the

linear and Gaussian case that has just been described. It explicitly determines the expectation and covariance matrix of the (Gaussian) conditional probability distribution, and evolves those quantities in time, updating these with new observations as they become available. EnKF, contrary to standard KF, evolves an ensemble of points in state space. One advantage is that it can be readily, if empirically, implemented on nonlinear dynamics. On the other hand, it keeps the same linear Gaussian procedure as KF

for updating the current uncertainty with new observations. EnKF exists in many variants and, even with





ensemble sizes of relatively small size O(10-100), produces results of high quality. It has now become, together with variational assimilation, one of the two most powerful algorithms used for assimilation in large dimension geophysical fluid applications.

Concerning the Bayesian properties of EnKF, Le Gland et al. (2009) have proven that, in the case of
linear dynamics and in the limit of infinite ensemble size, EnKF achieves Bayesian estimation, in that it determines the exact (Gaussian) conditional probability distribution. In the case of nonlinear dynamics, EnKF has a limiting probability distribution, which is not in general the Bayesian conditional distribution.

Contrary to EnKF, which was from the start developed for geophysical applications (but has since extended to other fields), *Particle Filters* (*PF*) have been developed totally independently of such ap-
plications. They are based on general Bayesian principles, and are thus independent of any hypothesis of linearity or Gaussianity (see Doucet et al. (2000) for more details). Like the EnKF, they evolve an ensemble of (usually weighted) points in state space, and update them with new observations as these become available. They exist in numerous variants, many of which have been mathematically proven to achieve Bayesianity in the limit of infinite ensemble size (but not, to the authors' knowledge, in the case
of finite ensemble size) . They are actively studied in the context of geophysical applications as presented in van Leeuwen (2009), but have not at this stage been operationally implemented on large dimension meteorological or oceanographical models.

There exist at least two other algorithms that can be utilised to build a sample of a given probability distribution. The first one is the *acceptance-rejection* algorithm described in Miller et al. (1999). The other
one is the *Metropolis-Hastings* algorithm (Metropolis et al. (1953)), which itself possesses a number of variants Robert (2015). These algorithms can be very efficient in some circumstances, but it is not clear at this stage whether they could be successfully implemented in large dimension geophysical applications.

Coming back to the linear and Gaussian case, not only, as said above, is the (Gaussian) conditional
probability distribution explicitly known, but a simple algorithm exists for determination of independent realizations of that distribution. In succinct terms, perturb additively the data according to their own error probability distribution, and perform the assimilation for the perturbed data. Repetition of this procedure on successive sets of independently perturbed data produces a Monte-Carlo sample of the Bayesian posterior distribution.
The present work is devoted to the study of that algorithm, and of its properties as a Bayesian estimator, in nonlinear and/or non-Gaussian cases. Systematic experiments are performed on two low-dimensional chaotic toy models, namely the Lorenz (1996) model and the Kuramoto-Sivashinsky equation (Kuramuto and Tsuzuki, 1975, 1976). Variational assimilation, which produces the Bayesian expectation in the linear



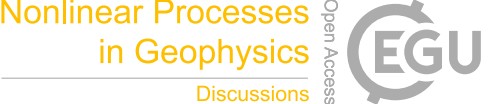

and Gaussian case, and is routinely, and empirically, implemented in nonlinear situations in operational

meteorology, is used for estimating the state vector for given (perturbed) data. The algorithm is therefore called *Ensemble Variational Assimilation*, abbreviated in *EnsVAR*.

This algorithm is not new. There exist actually a rather large number of algorithms for assimilation that are variational (at least partially) and build (at least at some stage) an ensemble of estimates of the state of the observed system. A review of those algorithms has been recently given by Bannister (2017).

Most of these algorithms are actually different from the one that is considered here. They have not been defined with the explicit purpose of achieving Bayesian estimation, and are not usually evaluated in that perspective.

On the other hand, Ensemble Variational Assimilation, as defined here, is actually routinely used at the European Centre for Medium-Range Weather Forecasts (ECMWF) (Isaksen et al. (2010)) in the defini-

tion of the initial conditions of ensemble forecasts. It is also used, both at ECMWF and at Météo-France (see respectively Bonavita et al. (2016) and Berre et al. (2015)), under the name Ensemble of Data Assimilations (EDA), for defining the background error covariance matrix of the Variational Assimilation system.

ECMWF, in its latest reanalysis project ERA 5 (Hersbach and Dee (2016)) uses a low resolution En-

semble of Data Assimilations system in order to estimate the uncertainty on the analysis. And, while the present papers were finalized, the authors became aware of a paper by Bardsley et al. (2014). These authors introduce a method, which they call Randomize-then-Optimize (RTO), which produces an ensemble of estimates through independent minimizations that take nonlinearity into account through the Jacobian of the model.

None of the above ensemble methods seems however to have been systematically and objectively evaluated as a probabilistic estimator. That is precisely the object of the present two papers.

The first of these is devoted to the exactly linear and weakly nonlinear cases, and the second to the fully nonlinear case. In this first one, the next Section describes in detail the EnsVAR algorithm, as well as the experimental set-up that is to be used in both parts of the work. The Section that follows then

describes the statistical tests to be used for objectively assessing EnsVAR as a probabilistic estimator. EnsVAR is implemented in Section 4, for reference, in an exactly linear and Gaussian case in which theory says it achieves exact Bayesian estimation. It is implemented in Section 5 on the nonlinear Lorenz system, over a relatively short assimilation window (5 days), over which the tangent linear approximation remains basically valid and the performance of the algorithm is shown not to be significantly altered.

Comparison is made in Section 6 with two standard algorithms for EnKF and PF. Experiments performed on the Kuramoto-Sivashinsky equation are succinctly presented in Section 7. Partial conclusions, valid


for weakly nonlinear case, are drawn in Section 8.

The second Part is devoted to the fully nonlinear situation, in which EnsVAR is implemented over assimilation windows for which the tangent linear approximation is no longer valid. Good performance

is nevertheless achieved through the technique of *Quasi Static Variational Assimilation* (*QSVA*), defined by Pires et al. (1996) and Järvinen et al. (1996). Comparison is made again with EnKF and PF.

The general conclusion of both Parts is that EnsVAR can produce good results which, in terms of Bayesian estimation, are at least as good as the results of EnKF and PF.

In the sequel of the paper we denote by $\mathcal{N}(\mathbf{m}, \boldsymbol{P})$ the multi-dimensional Gaussian probability distri-

bution with mean $\mathbf{m}$ and covariance matrix $\boldsymbol{P}$ (for a one-dimensional Gaussian probability distribution, we will use the similar notation $\mathcal{N}(m, r)$). $\mathbb{E}$ will denote statistical expectation, and $\mathbb{V}ar$ will denote variance.

## 2 The method of Ensemble Variational Assimilation

We assume the available data make up a vector $\mathbf{z}$, belonging to *data space* $\mathcal{D}$ with dimension $N_z$, of the

form

$$\mathbf{z} = \boldsymbol{\Gamma}\mathbf{x} + \boldsymbol{\zeta}, \tag{1}$$

In this expression, $\mathbf{x}$ is the unknown vector to be determined, belonging to *state space* $\mathcal{S}$ with dimension $N_x$, while $\boldsymbol{\Gamma}$ is a known linear operator from $\mathcal{S}$ into $\mathcal{D}$, called the *data operator* and represented by an $N_z \times N_x$ matrix. The $N_z$ vector $\boldsymbol{\zeta}$ is an 'error', assumed to be a realization of the Gaussian probability

distribution $\mathcal{N}(0, \boldsymbol{\Sigma})$ (in case the expectation $\mathbb{E}(\boldsymbol{\zeta})$ were non zero, but known, it would be necessary to first 'unbias' the data vector $\mathbf{z}$ by subtracting that expectation). It should be stressed that all available information about $\mathbf{x}$ is assumed to be included in the data vector $\mathbf{z}$. For instance, if one, or even several, Gaussian prior estimates $\mathcal{N}(\mathbf{x^b}, \mathbf{P^b})$ are available for $\mathbf{x}$, they must be introduced as subsets of $\mathbf{z}$, each with $N_x$ components, in the form

$\mathbf{x^b} = \mathbf{x} + \boldsymbol{\zeta^b}, \ \boldsymbol{\zeta^b} \sim \mathcal{N}(0, \mathbf{P^b}).$

In those conditions the Bayesian probability distribution $P(\mathbf{x}|\mathbf{z})$ for $\mathbf{x}$ conditioned by $\mathbf{z}$ is the Gaussian distribution $\mathcal{N}(\mathbf{x^a}, \mathbf{P^a})$ with

$$\begin{cases} \mathbf{x^a} = (\boldsymbol{\Gamma}^T \boldsymbol{\Sigma}^{-1} \boldsymbol{\Gamma})^{-1} \boldsymbol{\Gamma}^T \boldsymbol{\Sigma}^{-1} \mathbf{z} \\ \mathbf{P^a} = (\boldsymbol{\Gamma}^T \boldsymbol{\Sigma}^{-1} \boldsymbol{\Gamma})^{-1} \end{cases} \tag{2}$$

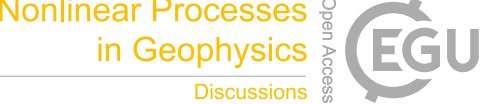



At first glance, the above equations seem to require the invertibility of the $N_z \times N_z$ matrix $\boldsymbol{\Sigma}$ and then, of the $N_x \times N_x$ matrix $\boldsymbol{\Gamma}^T \boldsymbol{\Sigma}^{-1} \boldsymbol{\Gamma}$. Without going into full details, the need for invertibility of $\boldsymbol{\Sigma}$ is only apparent, and invertibility of $\boldsymbol{\Gamma}^T \boldsymbol{\Sigma}^{-1} \boldsymbol{\Gamma}$ is equivalent to the condition that the data operator $\boldsymbol{\Gamma}$ is of rank $N_x$. This in turn means that the data vector $\mathbf{z}$ contains information on every component of $\mathbf{x}$. This condition is known as the *determinacy* condition. It implies that $N_z \geq N_x$. We will call $p = N_z - N_x$ the *degree of overdeterminacy* of the system.

The conditional expectation $\mathbf{x}^{\mathbf{a}}$ can be determined by minimizing the following scalar objective function defined on state space $\mathcal{S}$

$$\boldsymbol{\xi} \in \mathcal{S} \longrightarrow \mathcal{J}(\boldsymbol{\xi}) = \frac{1}{2} \left[ \boldsymbol{\Gamma}\boldsymbol{\xi} - \mathbf{z} \right]^T \boldsymbol{\Sigma}^{-1} \left[ \boldsymbol{\Gamma}\boldsymbol{\xi} - \mathbf{z} \right]. \tag{3}$$

In addition, the covariance matrix $\mathbf{P}^{\mathbf{a}}$ is equal to the inverse of the hessian of $\mathcal{J}$

$$\mathbf{P}^{\mathbf{a}} = \left[ \frac{\partial^2 \mathcal{J}}{\partial \boldsymbol{\xi}^2} \right]^{-1}. \tag{4}$$

In the case the error $\boldsymbol{\zeta}$, while still being random with expectation 0 and covariance matrix $\boldsymbol{\Sigma}$, is not Gaussian, the vector $\mathbf{x}^{\mathbf{a}}$ defined by Eq. (2) is not the conditional expectation of $\mathbf{x}$ for given $\mathbf{z}$, but only the least-variance linear estimate, or *Best Linear Unbiased Estimate* (*BLUE*), of $\mathbf{x}$ from $\mathbf{z}$. Similarly, the matrix $\mathbf{P}^{\mathbf{a}}$ is no longer the conditional covariance matrix of $\mathbf{x}$ for given $\mathbf{z}$, but the covariance matrix of the estimation error associated with the BLUE, averaged over all realizations of the error $\boldsymbol{\zeta}$.

Minimization of (3) can also been performed, at least in favorable circumstances, with a nonlinear data operator $\boldsymbol{\Gamma}$. This is what is done, heuristically but with undisputable usefulness, in meteorological and oceanographical *Variational Assimilation*. The latter is routinely implemented in a number of major meteorological centres, on nonlinear dynamical models with nonlinear observation operators. For more on minimization of objective functions of form (3) with nonlinear $\boldsymbol{\Gamma}$, see, *e.g.*, Chavent (2010).

Coming back to the linear and Gaussian case, consider the 'perturbed' data vector $\mathbf{z}' = \mathbf{z} + \boldsymbol{\zeta}'$, where the perturbation $\boldsymbol{\zeta}'$ has the same probability distribution $\mathcal{N}(0, \boldsymbol{\Sigma})$ as the error $\boldsymbol{\zeta}$. It is easily seen that the corresponding 'estimate'

$$\mathbf{x}^{\mathbf{a}\prime} = (\boldsymbol{\Gamma}^T \boldsymbol{\Sigma}^{-1} \boldsymbol{\Gamma})^{-1} \boldsymbol{\Gamma}^T \boldsymbol{\Sigma}^{-1} \mathbf{z}' \tag{5}$$

is distributed according to the Gaussian posterior distribution $\mathcal{N}(\mathbf{x}^a, \mathbf{P}^{\mathbf{a}})$ (Eq. 2). This defines a simple algorithm for obtaining a Monte-Carlo sample of that posterior distribution. Namely, perturb the data vector $\mathbf{z}$ according to its own error probability distribution, compute the corresponding 'estimate' (5), and repeat the same process with independent perturbations on $\mathbf{z}$.





That is the *Ensemble Variational Assimilation*, or *EnsVAR*, algorithm that is implemented below in nonlinear and non-Gaussian situations, the analogue of the estimate $\mathbf{x^{a\prime}}$ being computed by minimization
of form (3) . There is of course no reason to think that this approach will lead to Bayesian estimation, but it is interesting to study the properties of the ensembles thus obtained.

*Remark.* In the case when, the data operator $\mathbf{\Gamma}$ being linear, the error $\zeta$ in Eq. (1) is not Gaussian, the quantity $\mathbf{x^{a\prime}}$ defined by Eq. (5) has expectation $\mathbf{x^a}$ (BLUE) and covariance matrix $\mathbf{P^a}$ (see Isaksen et al., 2010). The probability distribution of the $\mathbf{x^{a\prime}}$ is not Bayesian, but it has the same expectation and
covariance matrix as the Bayesian distribution corresponding to a Gaussian $\zeta$.

All the experiments presented in this work are of the standard identical twin type, in which the 'observations' to be assimilated are extracted from a prior 'reference' integration of the assimilating model. And all experiments presented in this first Part are of the *strong constraint* variational assimilation type, in which the temporal sequence of states produced by the assimilation are constrained to satisfy exactly
the equations of the assimilating model.

That model, which will emanate from either the Lorenz or the Kuramoto-Sivashinsky equation, will be written as

$$\mathbf{x}_{t+1} = \mathfrak{M}(\mathbf{x}_t) \tag{6}$$

where $\mathbf{x}_t$ is the model state at time $t$, belonging to *model space* $\mathcal{M}$, with dimension $N$ (in the strong
constraint case considered in this first part, the model space $\mathcal{M}$ will be identical with the state space $\mathcal{S}$). For each model, a 'truth', or 'reference' run $\mathbf{x}_t^r$ has first been produced. A typical (strong constraint) experiment is as follows.

Choosing an assimilation window $[t_0, t_T]$ with length $T$ (it is mainly the parameter $T$ that will be varied in the experiments), synthetic observations are produced at successive times $(t_0 < t_1 < ... < t_k < $
$... < t_K = t_T)$, of the form

$$\mathbf{y}_k = H_k \mathbf{x}_k^r + \boldsymbol{\epsilon}_k \tag{7}$$

where $H_k$ is a linear observation operator, and $\boldsymbol{\epsilon}_k \sim \mathcal{N}(0, \mathbf{R}_k)$ is an 'observation error'. The $\boldsymbol{\epsilon}_k$'s are taken mutually independent.

The following process is then implemented $N_{ens}$ times $(iens = 1, \cdots, N_{ens})$
i/ Perturb the observations $\mathbf{y}_k, k = 0, \cdots, K$ according to

$$(\mathbf{y}_k^{iens})' = \mathbf{y}_k + \boldsymbol{\delta}_k \tag{8}$$

where $\boldsymbol{\delta}_k \sim \mathcal{N}(0, \mathbf{R}_k)$ is an independent realization of the same probability distribution that has produced $\boldsymbol{\epsilon}_k$. The notation $'$ stresses, as in Eq. (5), the "perturbed" character of $(\mathbf{y}_k^{iens})'$



ii/ Assimilate the perturbed observations $(\mathbf{y}_k^{iens})'$ by minimization of the following objective function

$$\boldsymbol{\xi}_0 \in \mathcal{M} \longrightarrow \mathcal{J}^{iens}(\boldsymbol{\xi}_0) = \frac{1}{2}\sum_{k=0}^{K}[H_k\boldsymbol{\xi}_k - (\mathbf{y}_k^{iens})']^T\,\mathbf{R}_k^{-1}\,[H_k\boldsymbol{\xi}_k - (\mathbf{y}_k^{iens})']. \quad (9)$$

where $\boldsymbol{\xi}_k$ is the value at time $t_k$ of the solution of the model (6) emanating from $\boldsymbol{\xi}_0$.

The objective function (9) is of type (3), the state space $\mathcal{S}$ being the model space $\mathcal{M}$ ($N = N_x$), and the data vector $\mathbf{z}$ consisting of the concatenation of the $K+1$ perturbed data vectors $(\mathbf{y}_k^{iens})'$.

The process i-ii, repeated $N_{ens}$ times, produces an ensemble of $N_{ens}$ model solutions over the assimi-
lation window $[t_0, t_T]$.

In the perspective taken here, it is not the properties of those individual solutions that matter the most, but the properties of the ensemble considered as a sample of a probability distribution.

The ensemble assimilation process, starting from Eq. (7), is then repeated over $N_{win}$ assimilation windows of length $T$ (taken sequentially along the true solution $\mathbf{x}_t^r$).

In variational assimilation as it is usually implemented, the objective function to be minimized contains a so-called *background* term at the initial time $t_0$ of the assimilation window. That term consists, together with an associated error covariance matrix, of a climatological estimate of the model state vector, or of a 'prior' estimate of that vector at time $t_0$ coming from assimilation of previous observations. An estimate of the state vector at $t_0$ is explicitly present in (9), in the form of the perturbed observation $(\mathbf{y}_0^{iens})'$. But
that is not a background term in the usual sense of the expression. In particular, no 'cycling' of any type is performed from one assimilation window to the next. The question of a possible cycling of ensemble variational assimilation will be discussed in Part II.

We sum up the description of the experimental procedure and define precisely the vocabulary to be used in the sequel. The output of one *experiment* consists of $N_{win}$ *ensemble variational assimilations*.
Each ensemble variational assimilation produces, through $N_{ens}$ minimizations of form (9), or *individual variational assimilations*, an ensemble of $N_{ens}$ model solutions corresponding to one set of observations $\mathbf{y}_k(k = 0, \cdots, K)$ over one assimilation window. These model solutions will be simply called the elements of the ensemble. The various experiments will differ through various parameters, and primarily the length $T$ of the assimilation windows.

The minimizations (9) are performed through an iterative Limited memory BFGS algorithm ( Nocedal and Wright (2006)), started from the observation $\mathbf{y}_0$ at time $t_0$ (which, as said below, is taken here as bearing on the entire state vector $\mathbf{x}_0^r$). Each step of the minimization algorithm requires the explicit knowledge of the local gradient of the objective function $\mathcal{J}^{iens}$ with respect to $\boldsymbol{\xi}_0$. That gradient is computed, as usual in variational assimilation, through the adjoint of the model (6). Unless specified
otherwise, the size of the assimilation ensembles will be $N_{ens} = 30$, and the number $N_{win}$ of ensemble



variational assimilations for one experiment will be equal to 9000.

## 3    The Validation Procedure

We recall first that, among all deterministic functions from data space into state space, the conditional expectation $\mathbf{z} \to \mathbb{E}(\mathbf{x}|\mathbf{z})$ minimizes the variance of the estimation error on $\mathbf{x}$.

What should ideally be done here is objectively assessing (if not on a case-to-case basis, at least in a statistical sense) whether the ensembles produced by EnsVAR are samples of the corresponding Bayesian probability distributions. In the present setting, where the probability distribution of the errors $\boldsymbol{\epsilon}_k$ in (7) is known, and where a prior probability distribution is also known, through the observation $\mathbf{y}_0$, for the state vector $\mathbf{x}_0$, one could in principle obtain a sample of the exact Bayesian probability distribution by

proceeding as follows. Through repeated independent realizations of the process defined by Eqs (6) and (7), build a sample of the joint probability distribution for the couple $(\mathbf{x}, \mathbf{z})$. That sample can then be read backwards for given $\mathbf{z}$ and, if large enough, will produce a useful sample estimate of the corresponding Bayesian probability distribution for $\mathbf{x}$. That would actually solve numerically the problem of Bayesian estimation. But it is clear that the sheer numerical cost of the whole process, which requires explicit

exploration of the joint space $(\mathbf{x}, \mathbf{z})$, makes this approach totally impossible in any realistic situation.

We have evaluated instead the weaker property of *reliability* (also called *calibration*). Reliability of a probabilistic estimation system (*i.e.* a system that produces probabilities for the quantities to be estimated) is statistical consistency between the predicted probabilities and the observed frequencies of occurrence. Consider a probability distribution $\pi$ (the words probability distribution must be taken here in the broadest

possible sense, meaning as well discrete probabilities for the occurrence of a binary or multi-outcome event, as continuous distributions for a one- or multi-dimensional random variable), and denote $\pi'(\pi)$ the distribution of the reality in the circumstances when $\pi$ has been predicted. Reliability is the property that, for any $\pi$, the distribution $\pi'(\pi)$ is equal to $\pi$.

Reliability can be objectively evaluated, provided a large enough verification sample is available.

Bayesianity clearly implies reliability. For any data vector z, the true state vector x is distributed according to the conditional probability distribution $P(\mathbf{x}|\mathbf{z})$, so that a probabilistic estimation system which always produce $P(\mathbf{x}|\mathbf{z})$ is reliable. The converse is clearly not true. A system which, ignoring the observations, always produces the climatological probability distribution for $\mathbf{x}$, will be reliable. It will however not be Bayesian (at least if, as one can reasonably hope, the observations bring more than climatological

information on the state of the system).

Another desirable property of a probabilistic estimation system, although not directly related to bayesian-



ity, is *resolution* (also called *sharpness*). It is the capacity of the system to *a priori* distinguish between different outcomes. For instance, a system which always predicts climatological probability distribution is perfectly reliable, but has no resolution. Resolution, like reliability, can be objectively evaluated if a

large enough verification sample is available.

We will use several standard diagnostic tools for validation of our results. We first note that the error in the mean of the predicted ensembles is itself a measure of resolution. The smaller that error, the higher the capacity of the system to *a priori* distinguish between different outcomes. Concerning reliability, the classical rank histogram and the *Reduced Centred Random Variable* (*RCRV*) (the latter is described

in Appendix A) are (non equivalent) measures of the reliability of probabilistic prediction of a scalar variable. The reliability diagram and the associated Brier score are relative to probabilistic prediction of a binary event. The Brier score decomposes into two parts, which measure respectively the reliability and the resolution of the prediction. The definition used here for those components is given in Appendix A (equations A4 and A5 respectively). Both scores are positive, and are negatively oriented, so that

perfect reliability and resolution are achieved when the corresponding scores take the value 0. For more on these diagnostics and, more generally, on objective validation of probabilistic estimation systems, see, *e.g.*, chapter 8 of the book by Wilks (2011), and the papers by Talagrand et al. (1997) and Candille and Talagrand (2005).

## 4  Numerical results: the linear case

We present in this section results obtained in an exactly linear and Gaussian case, in which theory says that EnsVAR must produce an exact Monte-Carlo Bayesian sample. These results are to be used as benchmark for the evaluation of later results. The numerical model (6) is obtained by linearizing the non-linear Lorenz model, which describes the space-time evolution of a scalar variable denoted $x$, about one particular solution (the Lorenz model will be described and discussed in more detail in the next Section,

see Eq. 12 below). The model space dimension $N$ is equal to 40. The length $T$ of the assimilation windows is 5 days, which covers $N_t = 20$ timesteps. The complete state vector ($H_k = I$ in Eq. 7) is observed every 0.5 day ($K = 10$). The data vector $\mathbf{z}$ has therefore dimension $(K+1)N = 440$. The observation errors are Gaussian, spatially uncorrelated, with constant standard deviation $\sigma = 0.1$ ($\mathbf{R}_k = \sigma^2\mathbf{I}, \forall k$) (however, because of the linearity, the absolute amplitude of those errors must have no impact).

Since conditions for exact Bayesianity are verified, any deviation in the results from exact reliability can be due to only the finiteness $N_{ens}$ of the ensembles (except for the rank histogram, which takes that finiteness into account), the finiteness $N_{win}$ of the validation sample or numerical effects (such as





resulting, for instance, from incomplete minimization or round-off errors).

Figure 1 shows the root-mean-square errors from the truth along the assimilation window, averaged at
each time over all grid points and all realizations. The upper (blue) curve shows the average error in the
individual minimizing solutions of $\mathcal{J}^{iens}$ (Eq. 9). The lower (red) curve shows the error in the mean of
the individual ensembles, while the green curve shows the error in the fields obtained in minimizations
performed with the raw unperturbed observations $\mathbf{y}_k$ (Eq. 7).

All errors are smaller than the observation error (horizontal dash-dotted line). The estimation errors
are largest at both ends of the assimilation window, and smallest at some intermediate time. As known,
and already discussed by various authors (Pires et al., 1996, Trevisan et al., 2010) , this is due to the
fact that the error along the stable components of the flow decreases over the assimilation window, while
the error along the unstable components increases. The ratio between the values on the blue and green
curves, averaged over the whole assimilation window, is equal to 1.414. This is close to $\sqrt{2}$ as can be
expected from the linearity of the process and the perturbation procedure defined by Equations (7-8)
(actually, it can be noted that the value $\sqrt{2}$ is itself, independently of any linearity, a test for reliability,
since the standard deviation of the difference between two independent realizations of a random variable
must be equal to $\sqrt{2}$ times the standard deviation of the variable itself). The green curve corresponds to
the expectation of (what must be) the Bayesian probability distribution, while the red curve corresponds
to a sample expectation, computed over $N_{ens}$ elements. The latter expectation is therefore not, as can be
seen on the figure, as accurate an estimate of the truth. The relative difference must be about $\frac{1}{2N_{ens}} \approx$
0.017. This is the value obtained here.

For a reliable system, the Reduced Centred Random Variable (RCRV), which we denote $s$, must have
expectation 0 and variance 1 (see Appendix A). The sample values, computed over all grid points, times
and assimilation windows (which amounts to a set of size $N_x \cdot (N_t + 1) \cdot N_{win} = 7.56.10^6$), are $\mathbb{E}(s) =$
0.0035 and $\mathbb{V}ar(s) = 1.00$.

Figure 2 shows other diagnostics of the statistical performance of the system, performed again over all
$7.56.10^6$ individual ensembles in the experiment. The top left panel is the rank histogram. The top right
panel is the reliability diagram relative to the event $\{x > 1.14\}$, which occurs with frequency 0.32 (black
horizontal dashed-dotted line in the diagram). Both panels visually show high reliability (flatness for the
histogram, closeness to the diagonal for the reliability diagram), although that reliability is obviously not
perfect. More accurate quantitative diagnostics are given by the lower panel, which shows, as functions
of the threshold $\tau$, the two components (reliability and resolution, see equations A4 and A5 respectively)
of the Brier score for the events $\{x > \tau\}$ . The reliability component is about $10^{-3}$, the resolution com-
ponent is about $5.10^{-2}$. A further diagnostic has been made by comparison with an experiment in which





the validating truth has been obtained, for each of the $N_{win}$ windows, from an additional independent $(N_{ens} + 1)st$ variational assimilation. That procedure is by construction perfectly reliable, and any difference with Figure 2 could result only from the fact that the validating truth is not defined by the same process. The reliability (not shown) is very slightly improved in comparison with Figure 2 (this could be

possibly due to a lack of full convergence of the minimizations). The resolution is not modified.

It is known that the minimum $\mathcal{J}_{min} = \mathcal{J}(\mathbf{x}^a)$ of the objective function (3) takes on average the value

$$\mathbb{E}(\mathcal{J}_{min}) = \frac{p}{2}, \tag{10}$$

where $p = N_z - N_x$ has been defined as the degree of overdeterminacy of the minimization. This result is

true under the only condition that the operator $\mathbf{\Gamma}$ is linear, and that the error $\boldsymbol{\zeta}$ in Eq. (1) has expectation $0$ and the covariance matrix $\boldsymbol{\Sigma}$ used in the objective function (3). It is independent of whether $\boldsymbol{\zeta}$ is Gaussian or not. But when $\boldsymbol{\zeta}$ is Gaussian, the quantity $2\mathcal{J}_{min}$ follows a $\chi^2$-probability distribution of order $p$ (for that reason, condition (10) is often called the $\chi^2$-condition, although it is verified in circumstances where $2\mathcal{J}_{min}$ does not follow a $\chi^2$-distribution). As a consequence, the minimum $\mathcal{J}_{min}$ has standard deviation

$$\boldsymbol{\sigma}(\mathcal{J}_{min}) = \sqrt{p/2}. \tag{11}$$

In the present case, $N_x = 40$ and $N_z = (K + 1)N_x = 440$, so that $p/2 = 200$ and $\sqrt{p/2} \approx 14.14$.

The histogram of the minima $\mathcal{J}_{min}$ (corrected for a multiplicative factor $1/2$ resulting from the additional perturbations (8)) is shown in Figure 3. The corresponding empirical expectation and standard deviation are 199.39 and 14.27 respectively, in agreement with Equations (10-11). It can be noted that,

as a consequence of the central limit theorem, the histogram in Figure 3 is in effect Gaussian. Indeed the value of negentropy, a measure of Gaussianity that will be defined in the next Section, is 0.0012.

For the theoretical conditions of exact Bayesianity considered here, reliability should be perfect, and should not be degraded when the information content of the observations decreases (through increased

observation error and/or degraded spatial and/or temporal resolution of the observations). Statistical resolution should, on the other hand, be degraded. Experiments have been performed to check this aspect (the exact experimental procedure is described in the next Section). The numerical results (not shown) are that both components of the Brier score are actually degraded, and can increase by one order of magnitude. The reliability component always remains much smaller than the resolution component, and

the degradation of the latter is much more systematic. This is in good agreement with the fact that the degradation of reliability can be due to only numerical effects, such as less efficient minimizations.

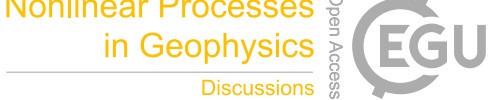



The above results, obtained in the case of exact theoretical Bayesianity, are going to serve as reference for the evaluation of EnsVAR in non-linear and non-Gaussian situations where Bayesianity does not hold.

## 5 Numerical results: the nonlinear case

The nonlinear Lorenz96 model (Lorenz (1996), Lorenz and Emanuel (1998)) reads

$$\frac{dx_j}{dt} = (x_{j+1} - x_{j-2})x_{j-1} - x_j + F, \qquad (12)$$

where $j = 1, \cdots, N$ represent the spatial coordinate ("longitude"), with cyclic boundary conditions. As in Lorenz (1996), we choose $N = 40$ and $F = 8$. For these values, the model is chaotic with 13 positive Lyapunov exponents, the largest of which has value $(2.5day)^{-1}$, where one day is equal to $0.44$ time unit
in Equation (12).

Except for the dynamical model, the experimental setup is fundamentally the same as in the linear case. In particular, the model time step 0.25 day, the observation frequency 0.5 day, and the values $N_{ens} = 30$ and $N_{win} = 9000$ are the same. The observation error is uncorrelated in space and time, with constant variance $\sigma^2 = 0.4$ ($\mathbf{R}_k = \sigma^2 \mathbf{I}, \forall k$). The associated standard deviation $\sigma = 0.63$ is equal to $2\%$ of
the variability of the reference solution. We mention again that no cycling is present between succesive assimilation windows.

The results are shown on Figure 4. The top panels are relative to one particular assimilation window. In the left panel, where the horizontal coordinate is the spatial position $j$, the black dashed curve is the reference truth at the initial time of the assimilation window, the blue circles are the corresponding
observations, and the full red curves ($N_{ens} = 30$ of them) are the minimizing solutions at the same time. The right panel, where the horizontal coordinate is time along the assimilation window, shows the truth (dashed curve) and the $N_{ens}$ minimizing solutions (full red curves) at three different points in space. Both panels show that the minimizations reconstruct the truth with a high degree of accuracy.

The bottom panel, which shows error statistics accumulated over all assimilation windows, is in the
same format as figure 1 (note that, because of the different dynamics and observational error, the amplitude on the vertical axis is different from figure 1). The conclusions are qualitatively the same. The estimation error, which is smaller than the observational error, is maximum at both ends of the assimilation window, and minimum at some intermediate time. The ratio between the blue and red curves, equal on average to $1.41$, is close to the value $\sqrt{2}$, which is in itself an indication of reliability. But a significant
difference is that the green curve lies now above the red curve. One obtains a better approximation of the truth by taking the average of the $N_{ens}$ minimizing solutions than by performing an assimilation on the





raw observations (7). This is an obvious nonlinear effect .

The expectation and variance of the RCRV are respectively $\mathbb{E}(s) = 0.012$ and $\mathbb{V}ar(s) = 1.047$.

Figure 5, which is in the same format as Figure 2, shows similar diagnostics : rank histogram, reliability

diagram for the event $\{x < 1.0\}$, which occurs whith frequency $0.33$, and the two components of the Brier score for events of the form $\{x > \tau\}$. The general conclusion is the same as in the linear case. High level of reliability is achieved. Actually, the reliability component of the Brier score (bottom panel) is now decreased below $10^{-3}$. That 'improvement', in the present situation where exact Bayesianity cannot be expected, can only be due to better numerical conditioning than in the linear case. The resolution

component of the Brier score, on the other hand, is increased.

Figure 6 is relative to experiments in which the informative content of the observations, *i.e.* their temporal density, spatial density, spatial resolution and accuracy (top, middle and bottom p and accuracy (top, middle and bottom panels respectively), has been varied. Each panel shows the two components of the Brier score, in the same format as in the bottom panels of Figures 2 and 5 (but with more curves

corresponding to different informative contents). The reliability component (red curves) always remains significantly smaller than the resolution component (blue curves). With the exception of the reliability component in the top panel, both components are systematically degraded when the information content of the observations decreases. This is certainly to be expected for the resolution component, but not necessarily for the reliability component. The degradation of the latter is significantly larger than in

the linear case (not shown), where we concluded that it could be due only to degradation of numerical conditioning. The degradation of reliability in the lower two panels may therefore be due here to non-linearity. One noteworthy feature is that the degradation of the resolution scores, for the same total decrease of the number of observations, is much larger for decrease of spatial density than for decrease of temporal density (middle and top panels respectively). This point has not been explored further, but is

consistent with the top two panels of Figure 4, which suggest that the model fields are more correlated in time than in space (right and left panelsrespectively). Less information is therefore lost in degrading the temporal than the spatial density of observations.

Figure 7 shows the distribution of (half) the minima of the objective function (it contains the same information as Figure 3, in a different format). Most values are concentrated around the 'linear' value 200,

but a small number of values are present in the range 600-1000. Excluding these outliers, the expectation and standard deviation of the minima are 199.62 and 14.13 respectively. These values are actually in better agreement with the theoretical $\chi^2$ values (200 and 14.14) than the ones obtained above in the theoretically exact bayesian case (199.39 and 14.27). This again suggests better numerical conditioning for the nonlinear situation.



In view of previous results, in particular results obtained by Pires et al. (1996), a likely explanation for the presence of the larger minima in Figure 7 is the following. Owing to the nonlinearity of Eq. (12), and more precisely to the 'folding' which occurs in state space as a consequence of the chaotic character of the motion, the uncertainty on the initial state is distributed along a folded subset in state space. It occasionally happens that the minimum of the objective function falls in a secondary fold,

which corresponds to a larger value of the objective function. This aspect will be further discussed in the second Part of the paper. In any case, the presence of larger minima of the objective function is an obvious sign of nonlinearity.

Nonlinearity is also obvious in Figure 8, which shows, for one particular minimization, a cross-section of the objective function between the starting point of the minimization and the minimum of the objective

function (black curve), and a parabola going through the starting point and having the same minimum (red curve). The two curves are distinctly different, while they would be identical in a linear case.

We have evaluated the Gaussian character of the ensembles produced by the assimilation by computing their *negentropy*. The negentropy of a probability distribution is the Kullback-Leibler divergence of that distribution with respect to the Gaussian distribution with the same expectation and variance (see Ap-

pendix B). The negentropy is positive and is equal to 0 for exact Gaussianity. The mean negentropy of the ensembles is here $\approx 10^{-3}$, indicating closeness to Gaussianity (for a reference, the empirical negentropy of a 30-element random Gaussian sample is $\approx 10^{-6}$ ). Although nonlinearity is present in the whole process, EnsVAR produces ensembles that are close to Gaussianity.

Experiments have been performed in which the observational error, instead of being Gaussian, has been

taken to follow a Laplace distribution (with still the same variance $\sigma^2 = 0.4$). No significant difference has been observed in the results in comparison with the Gaussian case. This suggests that the Gaussian character of the observational error is not critical for the conclusions obtained above.

## 6  Comparison with Ensemble Kalman Filter and Particle Filter

We present in this Section comparison with results obtained with the Ensemble Kalman Filter (EnKF)

and the Particle Filter (PF). As used here, those filters are sequential in time. Fair comparison is therefore possible only at the end of the assimilation window. Figure 9 shows the diagnostics obtained from EnsVAR at the end of the window (the top left panel, identical with the top right panel of Figure 4, is included for easy comparison with the figures that will follow). Comparison with Figure 5 shows that the reliability (as measured by the rank histogram, the reliability diagram and the reliability component of

the Brier score) is significantly degraded. It has been verified (not shown) that this degradation is mostly



due, not to a really degraded performance at the end of the window, but to the use of a smaller validation sample (by a factor of $N_t + 1 = 21$, which leads to a sample with size $3.6.10^5$).

Figure 10, which is in the same format as Figure 9, shows the same diagnostics for the EnKF. The algorithm used is the one described by Evensen (2003). In particular, observations have been perturbed randomly, for updating the background ensembles, according to the probability distribution of the obser-
vation errors. Spatial localization of the background error covariance matrix has been implemented by Schur-multiplying the sample covariance matrix by a squared exponential kernel with lengthscale 12.0 (the positive definiteness of the periodic kernel has been ensured by removing its negative Fourier components). And multiplicative inflation with factor r = 1.001, has been applied, as in Anderson and Anderson
(1999), on the ensemble after each analysis.

Comparison with Figure 9 shows that the individual ensembles, after a 'warm up' period, tend to remain more dispersed than in EnsVAR (top left panel). Reliability, as measured by the reliability diagram and the Brier score, is similar to what it is in Figure 9. But it is significantly degraded as evaluated by the rank histogram. The ensembles, although they have larger absolute dispersion than in EnsVAR, tend to 'miss'
reality more often.

Figure 11 (again in the same format as Figure 9) shows the same diagnostics for a Particle Filter. The algorithm used here is the Sampling Importance Particle Filter presented in Arulampalam et al. (2002). Comparison with Figure 10 shows first that the individual ensembles are still more dispersed than in EnKF (top left panel). It also shows a slight degradation of the reliability component of the Brier score
(and, incidentally, a significant degradation of the resolution component), but no visible difference on the reliability diagram. Concerning the rank histogram, PF produces unequally weighted particles, and the standard histogram could not be used. A histogram has been built instead on the quantiles defined by the weights of the particles. This shows, as for EnKF, a significant global deviation from reality.

The left column of Table 1 shows the mean root-mean square error in the means of the ensembles as
obtained from the three algorithms. The performance of EnsVAR and EnKF (0.22 and 0.24) is comparable by that measure, while the performance of PF is significantly worse (0.76).

Figures 12 to 14 are relative to ensemble forecasts performed, for each of the three assimilation algorithms, from the ensembles obtained at the end of the 5-day assimilations. They are in the same format as Figure 9, and show diagnostics at the end of 5-day forecasts. One can first observe that the dispersion of
individual forecasts (top left panels) increases, as can be expected, with the forecast range, but much less with the EnsVAR than with EnKF and PF. Reliability, as measured by the Brier score, is slightly degraded in all three algorithms with respect to the case of the assimilations. It is slightly worse for EnKF than for EnsVAR, and significantly worse for PF. Resolution is on the other hand significantly degraded in





| DA procedure / method | Assimilation | Forecasting |
|---|---|---|
| EnsVAR | 0.22 | 1.49 |
| EnKF | 0.24 | 1.67 |
| PF | 0.76 | 2.63 |

Table 1: RMS errors at the end of 5 days of assimilation (left column) and of 5 days of forecast (right column) for the three algorithms

all three algorithms. This is associated with the dispersion of ensembles, and corresponds to what could
be expected. Concerning the rank histograms, the histogram of EnsVAR, although still noisy, shows no
systematic sign of over- or underdispersion of the ensembles. The EnKF and PF histograms both present,
as before, a significant underdispersion.

Finally, the right column of Table 1 shows that RMS errors, which are of course now larger, still rank
comparatively in the same order as before, *i.e.* EnsVAR < EnKF < PF.

**7 The Kuramoto-Sivashinsky equation**

Similar experiments have been performed with the Kuramoto-Sivashinsky (K-S) equation. It is a one-
dimensional spatially periodic evolution equation, with an advective nonlinearity, a fourth-order dissipa-
tion term and a second-order anti-dissipative term. It reads

$$
\begin{cases}
\dfrac{\partial u}{\partial t} + \dfrac{\partial^4 u}{\partial x^4} + \dfrac{\partial^2 u}{\partial x^2} + u\dfrac{\partial u}{\partial x} = 0, \ \ x \in [0, L] \\[2mm]
\dfrac{\partial^i u}{\partial x^i}(x + L, t) = \dfrac{\partial^i u}{\partial x^i}(x, t) \text{ for } i = 0, 1, \cdots, 4, \ \forall t > 0 \\[2mm]
u(x, 0) = u_0(x)
\end{cases}
\tag{13}
$$

where the spatial period $L$ is a bifurcation parameter for the system. The K-S equation models pattern
formations in different physical contexts and is a paradigm of low-dimensional behavior in solutions to
partial differential equations. It arises as a model amplitude equation for inter-facial instabilities in many
physical contexts. It was originally derived by (Kuramuto and Tsuzuki, 1975, 1976) to model small





thermal diffusive instabilities in laminar flame fronts in two space dimensions. Equation (13) has been

used here with the value $L = 32\pi$ and has been discretized to 64 Fourier modes. In accordance with the calculations of Manneville (1985), we observe chaotic motion with 27 positive Lyapunov exponents, the largest one being $\lambda_{max} \approx 0.13$.

With $L = 32\pi$ and the initial condition

$$u(x,0) = \cos(\frac{x}{16})\left(1 + \sin(\frac{x}{16})\right)$$

(14)

The system (13) is known to be stiff. The stiffness is due to rapid exponential decay of some modes (the dissipative part), and to rapid oscillations of other modes (the dispersive part).

Figure 15, where the two panels are in the same format as Figure 1, shows the errors in the EnsVAR assimilations, in both a linearized (top panel) and a fully nonlinear case (bottom panel) cases. The length of the assimilation window, marked as 1 on the figure, is equal to $\dfrac{1}{\lambda_{max}} \approx 7.7$ in units of Equation (13),

*i.e.* a typical predictability time of the system. The shapes of the curves show that the KS equation has globally more stability and less instability than the Lorenz equation. The figure shows similar performance for the linear and nonlinear situation. Other results (not shown) are also qualitatively very similar to those obtained with the Lorenz equation: high reliability of the ensembles produced by EnsVAR, and slightly superior performance over EnKF and PF.

## 8    Summary and conclusions

Ensemble Variational Assimilation (EnsVAR) has been implemented on two small dimension non-linear chaotic toy models, as well as on linearized version of those models.

One specific goal of the paper was to stress what is in the authors' mind a critical aspect, namely to systematically evaluate ensembles produced by ensemble assimilation as probabilistic estimators. This

requires to consider these ensembles as defining probability distributions (instead of evaluating them principally, for instance, by the error in their mean).

In view of the impossibility of objectively validating the Bayesianity of ensembles, the weaker property of reliability has been evaluated instead. In the linear and Gaussian case, where theory says that EnsVAR is exactly Bayesian, the reliability of the ensembles produced by EnsVAR is high, but not numerically

perfect, showing the effect of sampling errors and, probably, of numerical conditioning.

In the nonlinear case, EnsVAR, implemented on temporal windows on the order of magnitude of the predictability time of the systems, shows as good (and in some cases slightly better) performance as in the exactly linear case. Comparison with Ensemble Kalman Filter (EnKF) and Particle Filter (PF) shows





EnsVAR is globally as good a statistical estimator as those two other algorithms.

On the other hand, EnsVar, at it has been implemented here, is numerically more costly than either EnKF or PF. And the specific algorithms used for the latter two methods may not be the most efficient. But it is worthwhile to evaluate EnsVAR in the more demanding conditions of stronger nonlinearity. That is the object of the second part of this work.

**Appendix A**

**Methods for Ensemble Evaluation**

This Appendix describes in some detail two of the scores that are used for evaluation of results in the paper, namely the *Reduced Centred Random Variable* (*RCRV*) and the reliability-resolution decomposition of the classical Brier score. Given a 'predicted' probability distribution for a scalar variable $x$ and a verifying observation $\xi$, the corresponding value of the Reduced Centred Random Variable is defined as

$$s \equiv \frac{\xi - \mu}{\sigma}, \tag{A1}$$

where $\mu$ and $\sigma$ are respectively the mean and the standard deviation of the predicted distribution. For a perfectly reliable prediction system, and over all realizations of the system, $s$, by the very definition of expectation and standard deviation, has expectation 0 and variance 1. This is true independently of whether or not the predicted distribution is always the same. An expectation of $s$ that is different from 0

means that the system is globally biased. If the expectation is equal to 0, a variance of $s$ that is smaller (resp. larger) than 1 is sign of global over- (resp. under-) dispersion of the predicted distribution. One can note that, contrary to the rank histogram, which is invariant in any monotonous one-to-one transformation on the variable $x$, the $RCRV$ is invariant only in a linear transformation.

We recall the *Brier score* for a binary event $\mathscr{E}$ is defined by

$$\mathbb{B} = \mathbb{E}\left[(p - p_0)^2\right] \tag{A2}$$

where $p$ is the probability predicted for the occurrence of $\mathscr{E}$ in a particular realization of the probabilistic prediction process, $p_0$ is the corresponding *verifying* observation ($p_0 = 1$ or 0 depending on whether $\mathscr{E}$ has been observed to occur or not), and $\mathbb{E}$ denotes the mean taken over all realizations of the process. Denoting by $p'(p)$, for any probability $p$, the frequency with which $\mathscr{E}$ is observed to occur in the

circumstances when $p$ has been predicted, $\mathbb{B}$ can be rewritten as



$$\mathbb{B} = \mathbb{E}\left[(p - p')^2\right] + \mathbb{E}\left[p'(1 - p')\right] \tag{A3}$$

The first term on the right-hand side, which measures the horizontal dispersion of the points on the reliability diagram about the diagonal, is a measure of reliability. The second term, which is a (negative) measure of the vertical dispersion of the points, is a measure of resolution (the larger the dispersion, the higher the resolution, and the smaller the second term on the right-hand side). It is those two terms, divided by the constant $p_c(1 - p_c)$, where $p_c = \mathbb{E}(p_0)$ is the overall observed frequency of occurrence of $\mathscr{E}$, that are taken in the present paper as measures of reliability and resolution

$$\mathbb{B}_{reli} = \frac{\mathbb{E}\left[(p - p')^2\right]}{p_c(1 - p_c)} \tag{A4}$$

$$\mathbb{B}_{reso} = \frac{\mathbb{E}\left[p'(1 - p')\right]}{p_c(1 - p_c)} \tag{A5}$$

Both measures are negatively oriented, and have $0$ as optimal value. $\mathbb{B}_{reli}$ is bounded above by $1/p_c(1 - p_c)$, while $\mathbb{B}_{reso}$ is bounded by $1$.

*Remark.* There exist other definitions of the reliability and resolution components of the Brier score. In particular, concerning resolution, the 'uncertainty' term $p_c(1 - p_c)$ (which depends on the particular event $\mathscr{E}$ under consideration) is often subtracted from the start from the raw score (A2). This leads to slightly different scores.

As said in the main text, more on the above diagnostics and, more generally, on objective validation of probabilistic estimation systems, can be found in, *e.g.*, chapter 8 of the book by Wilks (2011), or in the papers by Talagrand et al. (1997) and Candille and Talagrand (2005).

**Appendix B**

**Negentropy**

The negentropy of a probability distribution with density $f(y)$ is the Kullback-Leibler divergence, or relative entropy, of that distribution with respect to the Gaussian distribution with the same expectation and variance. Denoting by $f_{\mathrm{G}}(y)$ the density of that Gaussian distribution, the negentropy can be expressed as

$$N(f) = \int f(y) \ln\left[\frac{f(y)}{f_{\mathrm{G}}(y)}\right] dy \tag{B1}$$




The negentropy is always positive, and is equal to 0 if and only if density $f(y)$ is Gaussian. As examples, a Laplace distribution has negentropy 0.072, while the empirical negentropy of a 30-element random Gaussian sample is $\approx 10^{-6}$. In the case of small skewness $s$ and normalized kurtosis $k$, the negentropy can be approximated by

590
$$N(f) \approx \frac{1}{12}s^2 + \frac{1}{48}k^2 \qquad (B2)$$

It is this formula that has been used in the present paper.

*Acknowledgements.* This work has been supported by Agence Nationale de la Recherche, France, through the Prevassemble and Geo-Fluids projects, as well as by the programme Les enveloppes fluides et l'environnement of Institut national des sciences de l'Univers, Centre national de la recherche scientifique, Paris. The authors acknowledge fruit-
595 ful discussions with M. Bocquet and J. Brajard.

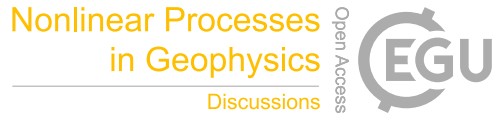

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





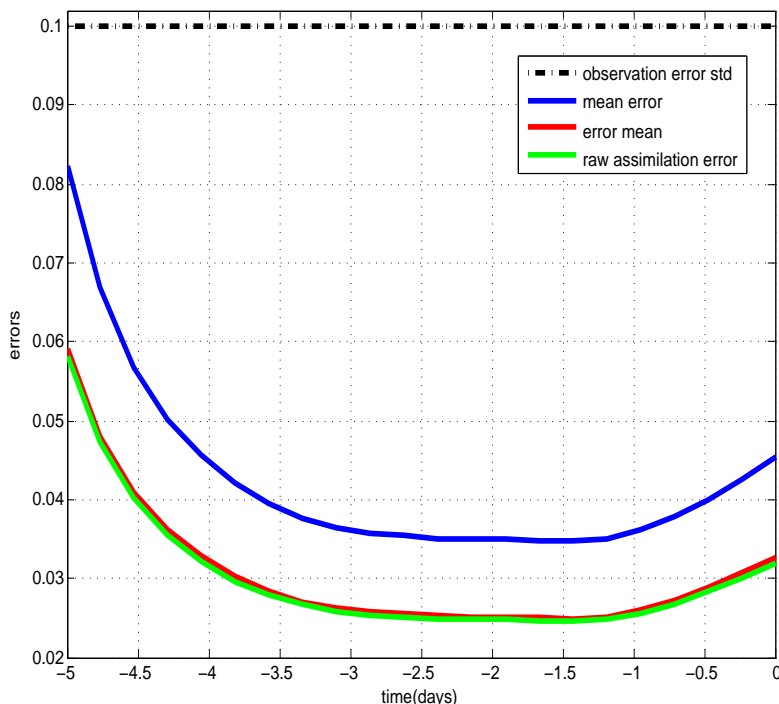

Fig. 1: Root-mean square errors from the truth as functions of time along the assimilation window (linear and Gaussian case). Blue curve : error in individual minimizations. Red curve : error in the means of the ensembles. Green curve : error in the assimilations performed with the unperturbed observations $\mathbf{y}_k$ (Eq. 7). Dashed horizontal curve : standard deviation of the observation error. Each point on the blue curve corresponds to an average over a sample of $N_x \cdot N_{win} \cdot N_{ens} = 1.08.10^7$ elements, each point on the red and green curves to an average over a sample of $N_x \cdot N_{win} = 3.6.10^5$ elements.




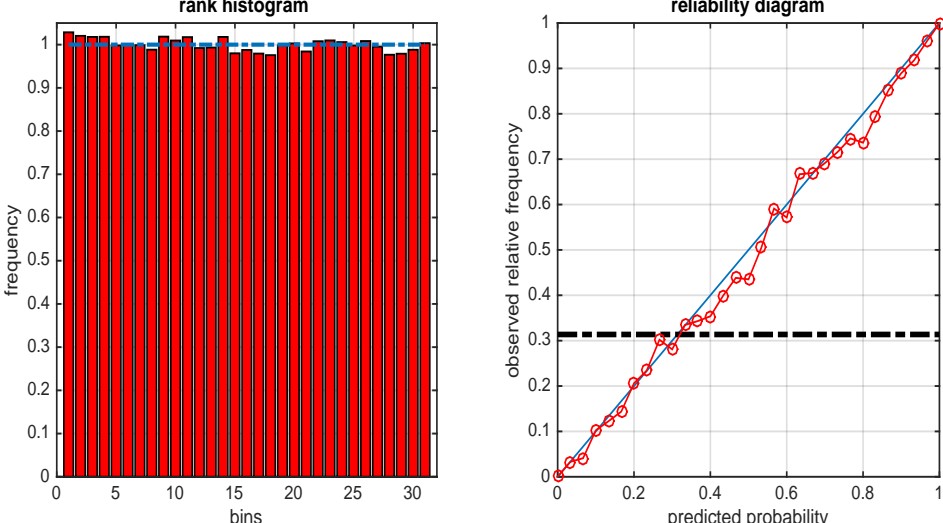

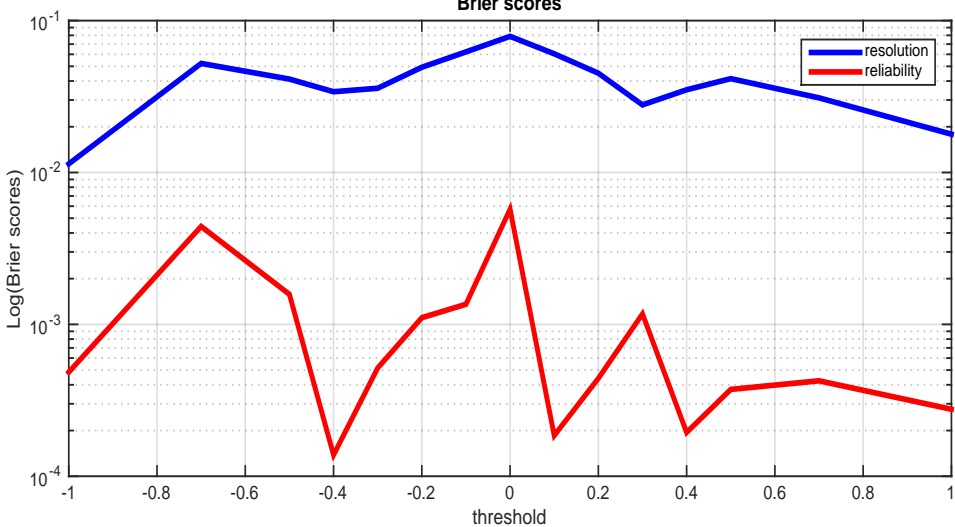

Fig. 2: Diagnostics of statistical performance (linear and Gaussian case). Top left: rank histogram for the model variable $x$. Top right : reliability diagramme for the event $\mathscr{E} = \{x > 1.14\}$ (black horizontal dot-dashed line : frequency of occurrence of the event). Bottom : variation with threshold $\tau$ of the reliability and resolution components of the Brier score for the events $\mathscr{E} = \{x > \tau\}$ (red and blue curves respectively, note the logarithmic scale on the vertical). The diagnostics have been computed over all gridpoints, timesteps and realizations, making up a sample of size $7.56.10^6$.





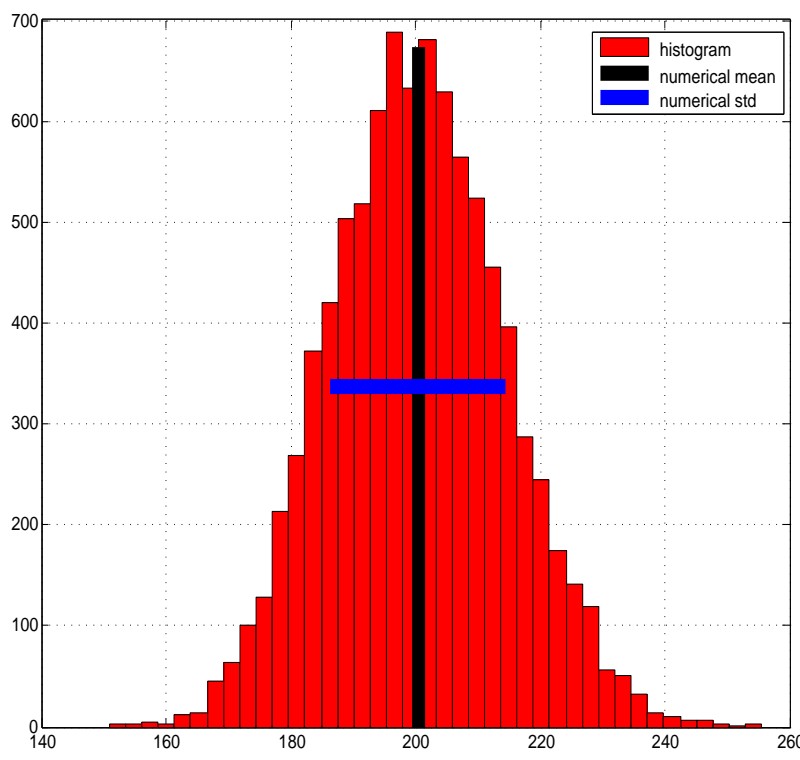

Fig. 3: Histogram of (half) the minima of the objective function (9), along with the corresponding mean (vertical black line) and standard deviation (horizontal blue line) (linear and Gaussian case).


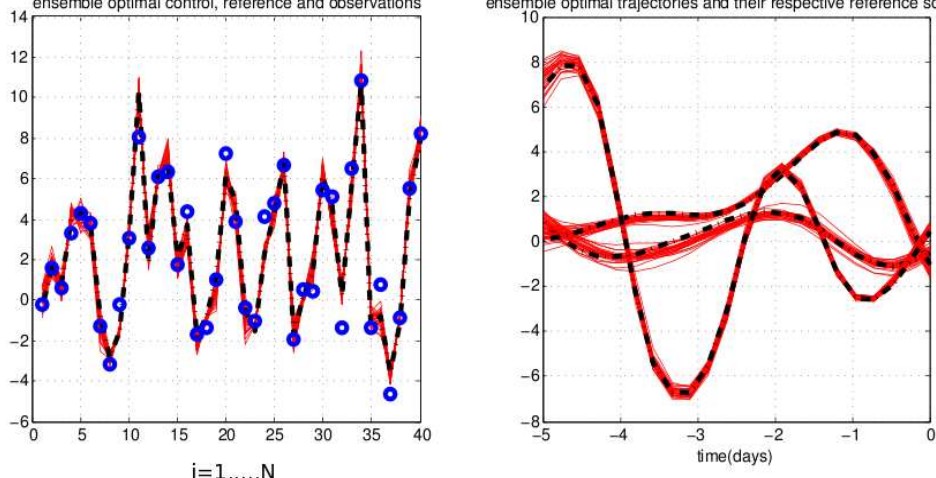

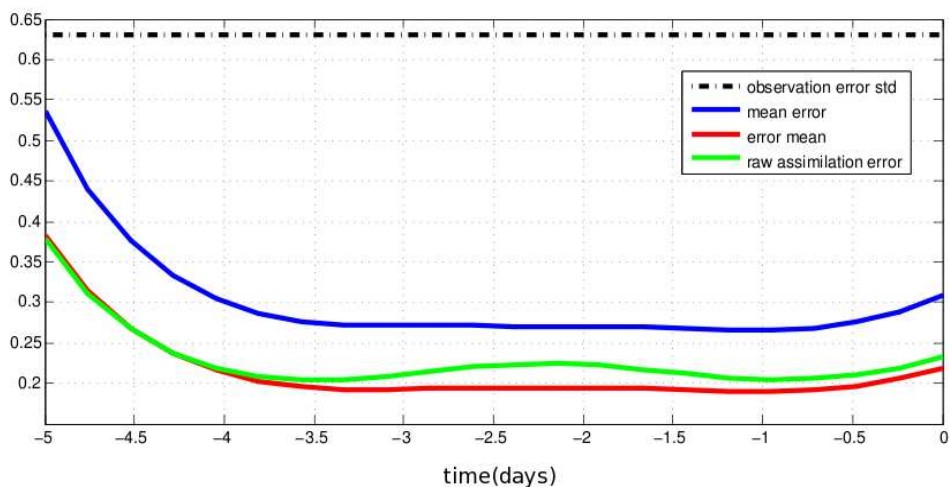

Fig. 4: Diagnostics relative to the non-linear and Gaussian case, with assimilation over 5 days. Top panels are relative to one particular assimilation window. Left (horizontal coordinate : spatial position $j$) : reference truth at the initial time of the assimilation window (black dashed curve), observations (blue circles), minimizing solutions (full red curves). Right (horizontal coordinate : time along the assimilation window): truth (dashed curve) and minimizing solutions (full red curves) at three points in space. Bottom panel : overall diagnostics of estimation errors (same format as in figure 1).





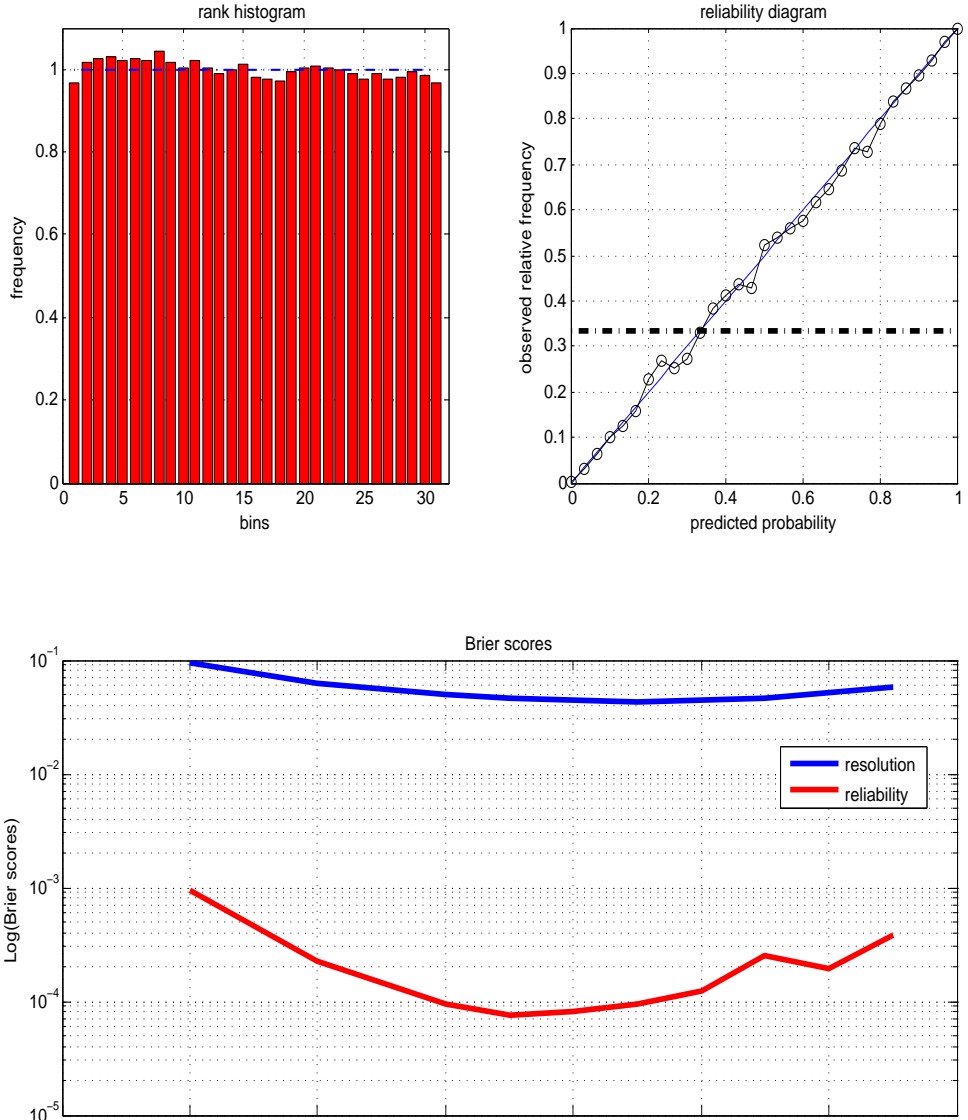

Fig. 5: Same as Figure 2, for the non-linear case (for the event $\mathscr{E} = \{x < 1.\}$ , which occurs with frequency 0.33, as concerns the reliability diagramme on the top right panel).

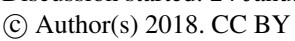


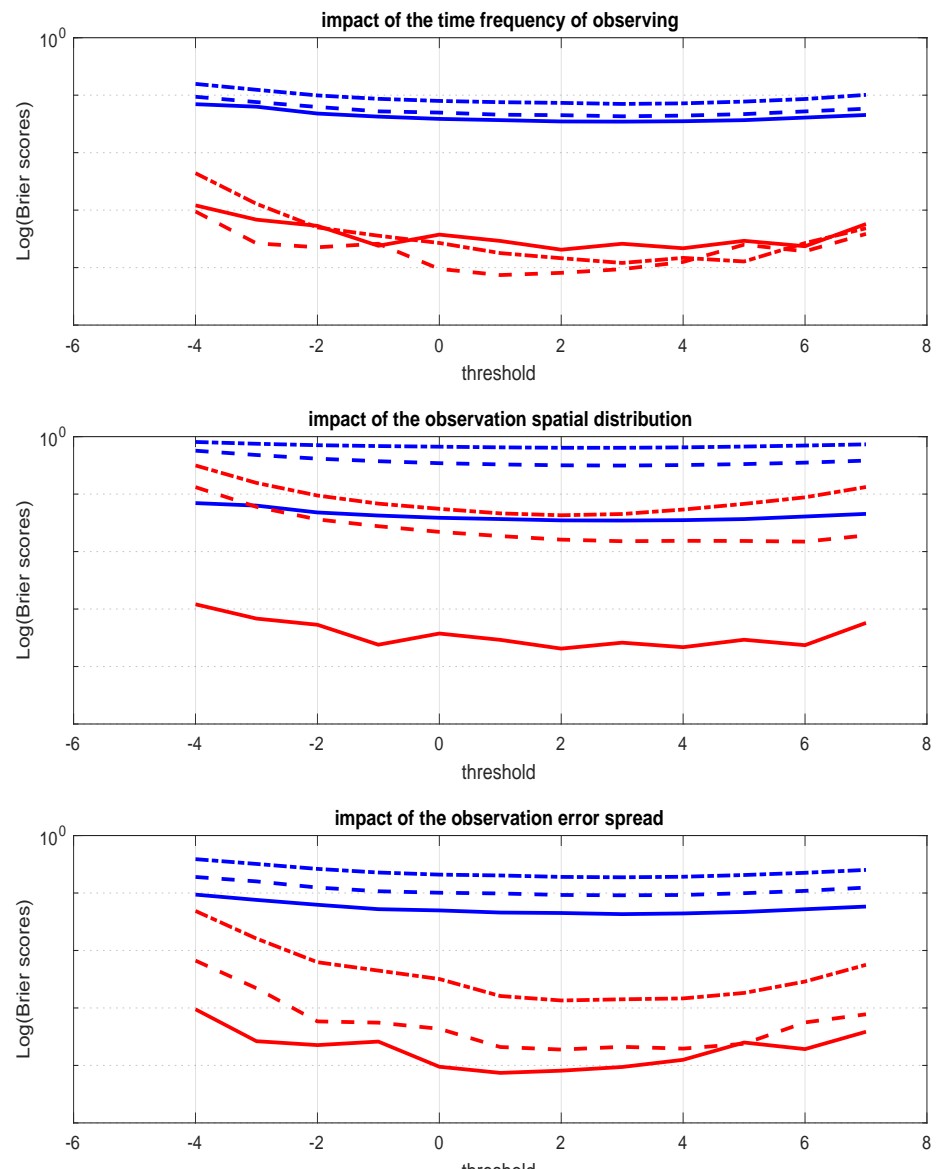

Fig. 6: Impact of the informative content of observations on the two components of the Brier score (non-linear case). The format of each panel is the same as the format of the bottom panels of Figures 2 and 5 (red and blue curves : reliability and resolution components respectively). Top panel : impact of the temporal density of the observations. Observations are performed every gridpoint, with error variance $\sigma^2 = 0.4$, and every time step (full curves), every second, and fourth timestep (dashed, and dash-dotted curves respectively). Middle panel : impact of the spatial density of the observations. Observations are performed every timestep, with error $\sigma^2 = 0.4$, and at every gridpoint (full curves), and every second and fourth gridpoint (dashed and dash-dotted curves respectively). Bottom panel : impact of the variance $\sigma^2$ of the observation error. Observations are performed every second timestep and at every gridpoint with observation error std $\sigma = \sqrt{0.4}, 2\sqrt{0.4}$, and $4\sqrt{0.4}$ (full, dashed and dash-dotted curves respectively.





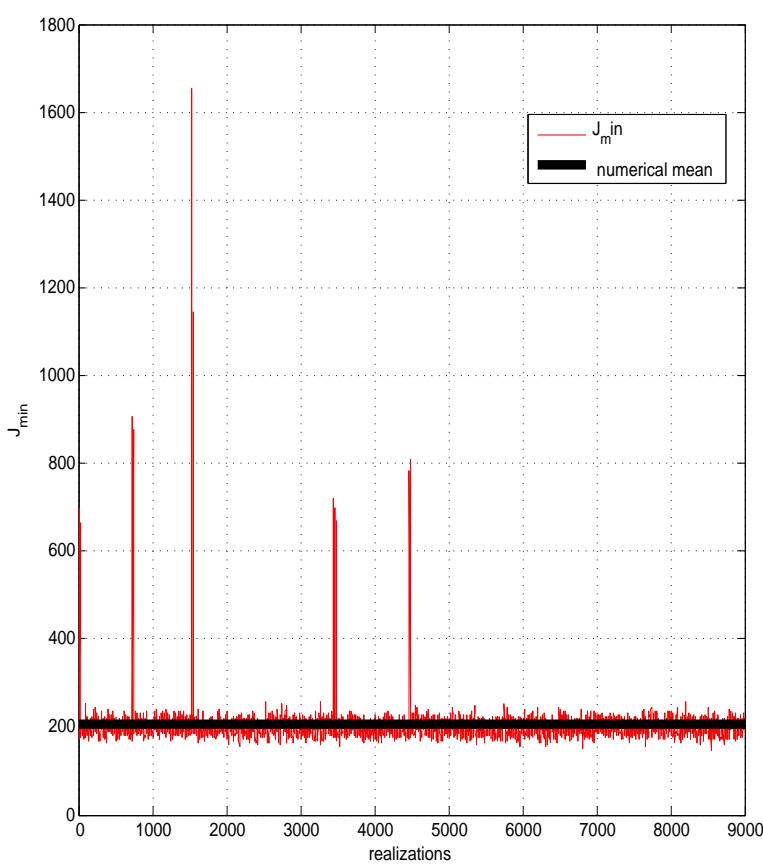

Fig. 7: Values of (half) the minima of the objective function for all realizations (non-linear case).



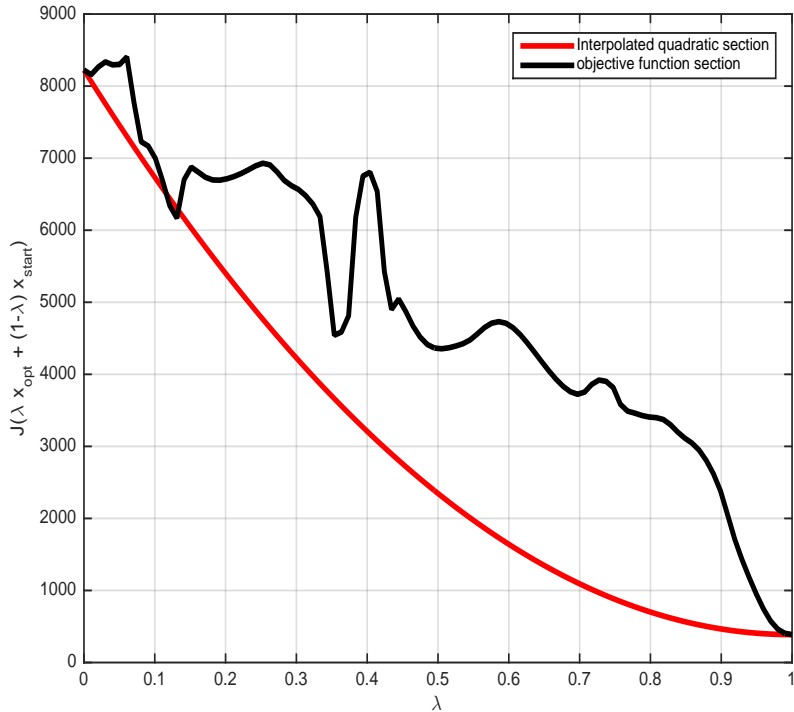

Fig. 8: Cross-section of the objective function $\mathcal{J}^{iens}$, for one particular minimization, between the starting point of the minimization and the minimum of $\mathcal{J}^{iens}$ (black curve). Parabola going through the starting point and having the same minimum (red curve).




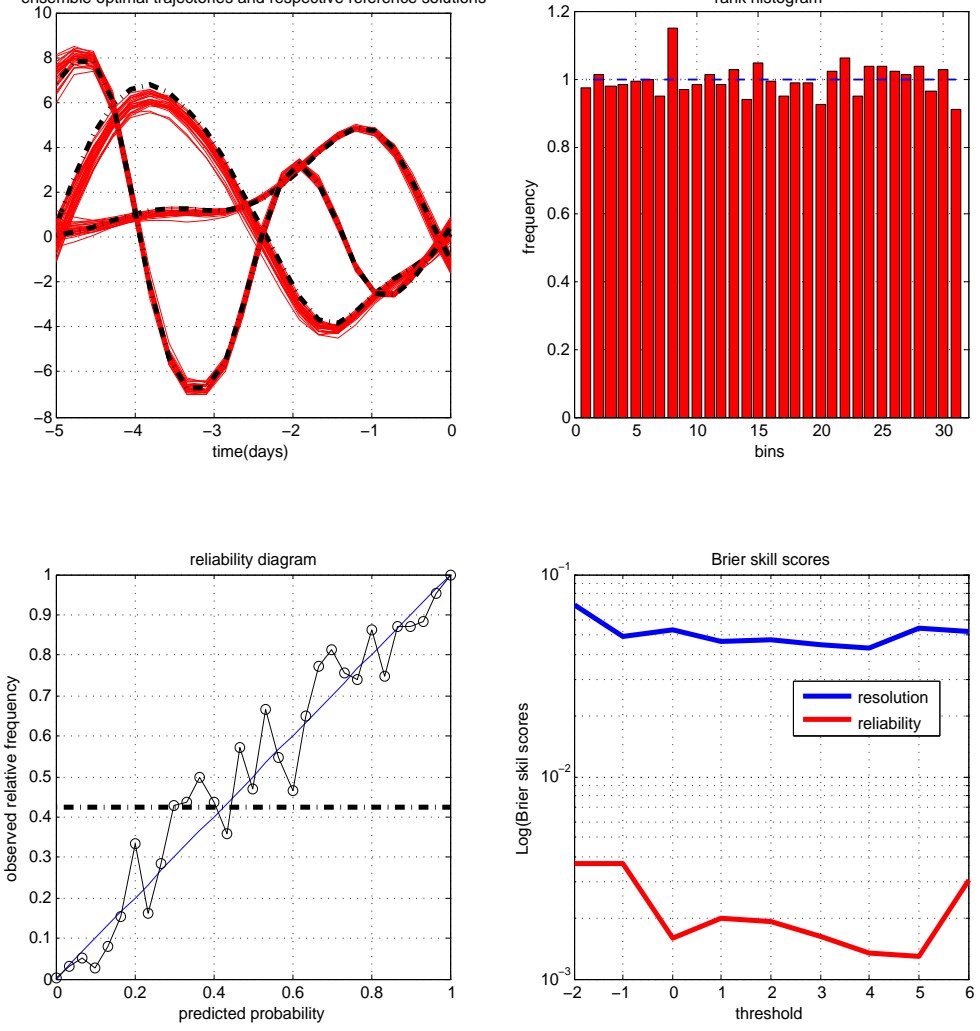

Fig. 9: Top left panel : Identical with the top right panel of Figure 4, repeated for comparison with figures that follow. The other panels show the same diagnostics as in Figure 5, but performed at the final time of the assimilation windows. Top right : rank histogram. Bottom left: reliability diagram for the event $\mathscr{E} = \{x > 1.33\}$, which occurs with frequency 0.42. Bottom right : components of the Brier score or the same event (same format as in the bottom panels of Figures 2 and 5.





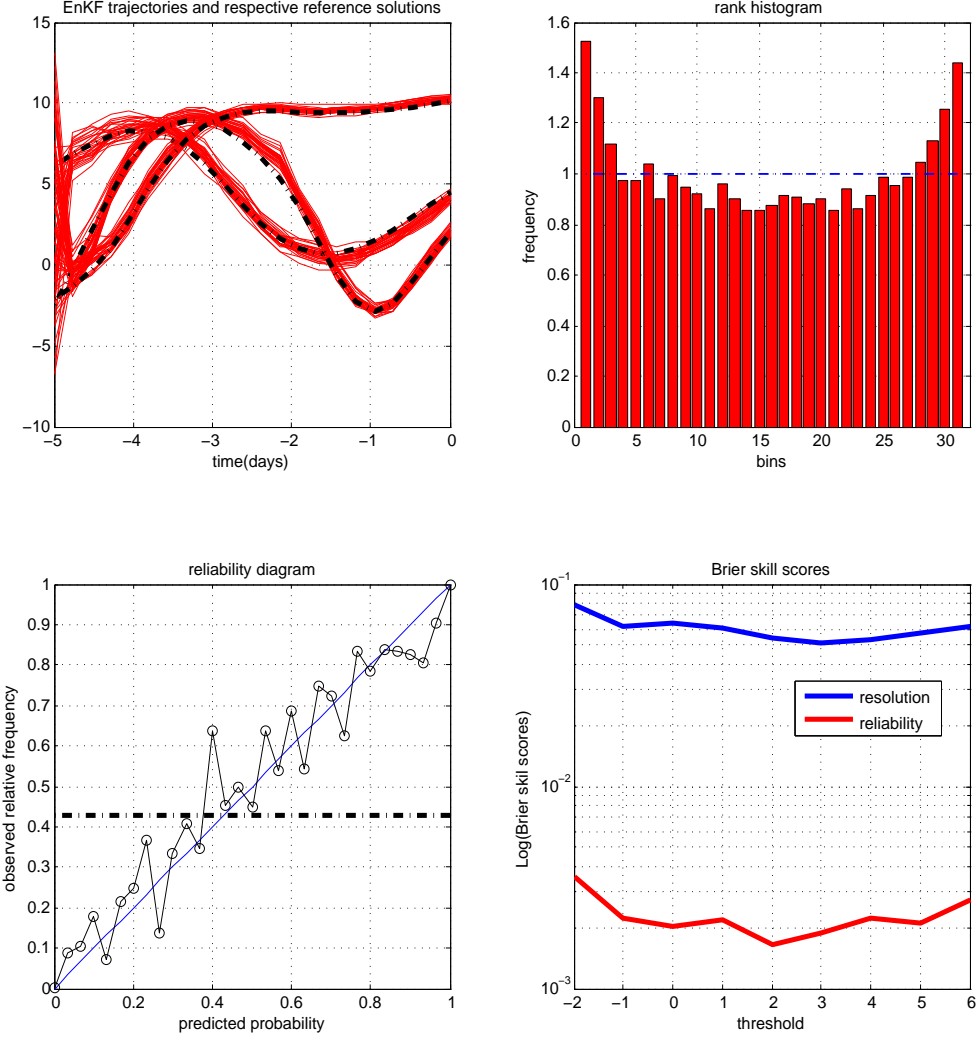

Fig. 10: Same as Figure 9, for the Ensemble Kalman Filter.

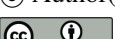


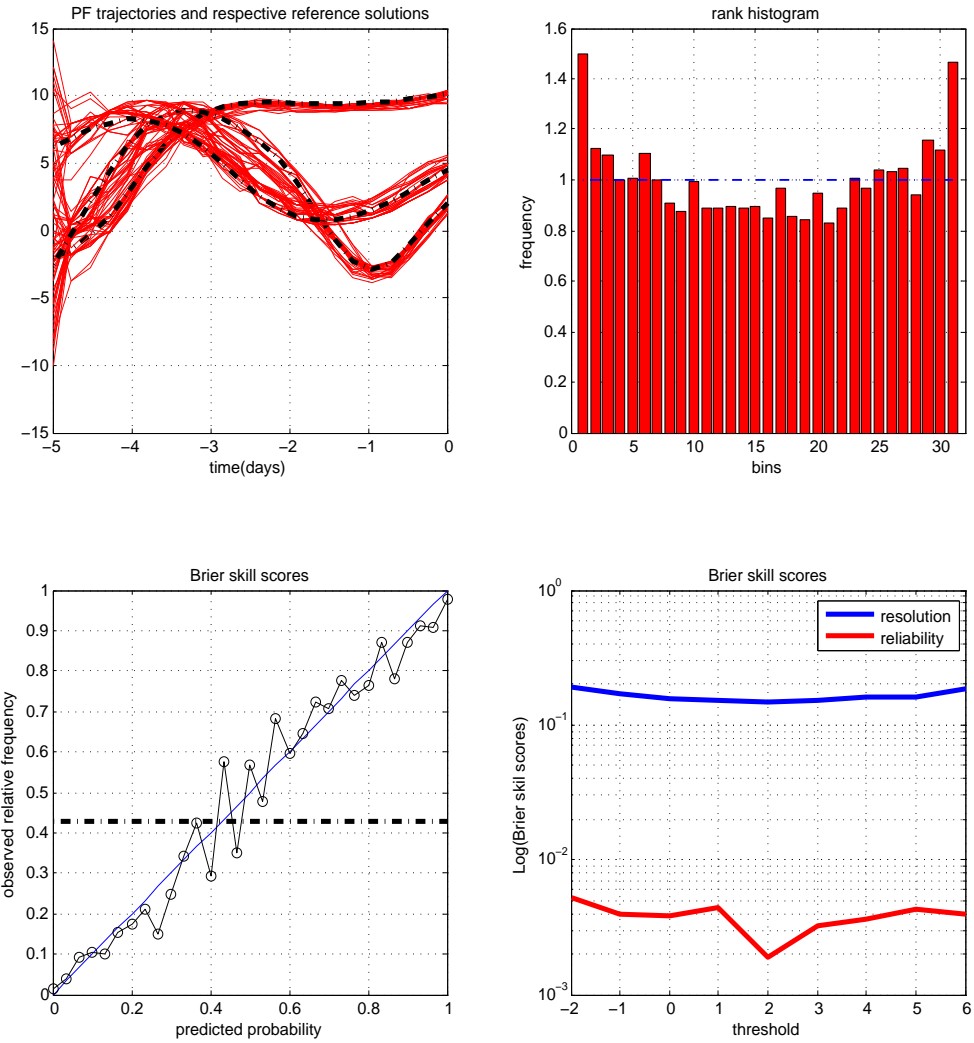

Fig. 11: Same as Figure 9, for the Particle Filter.




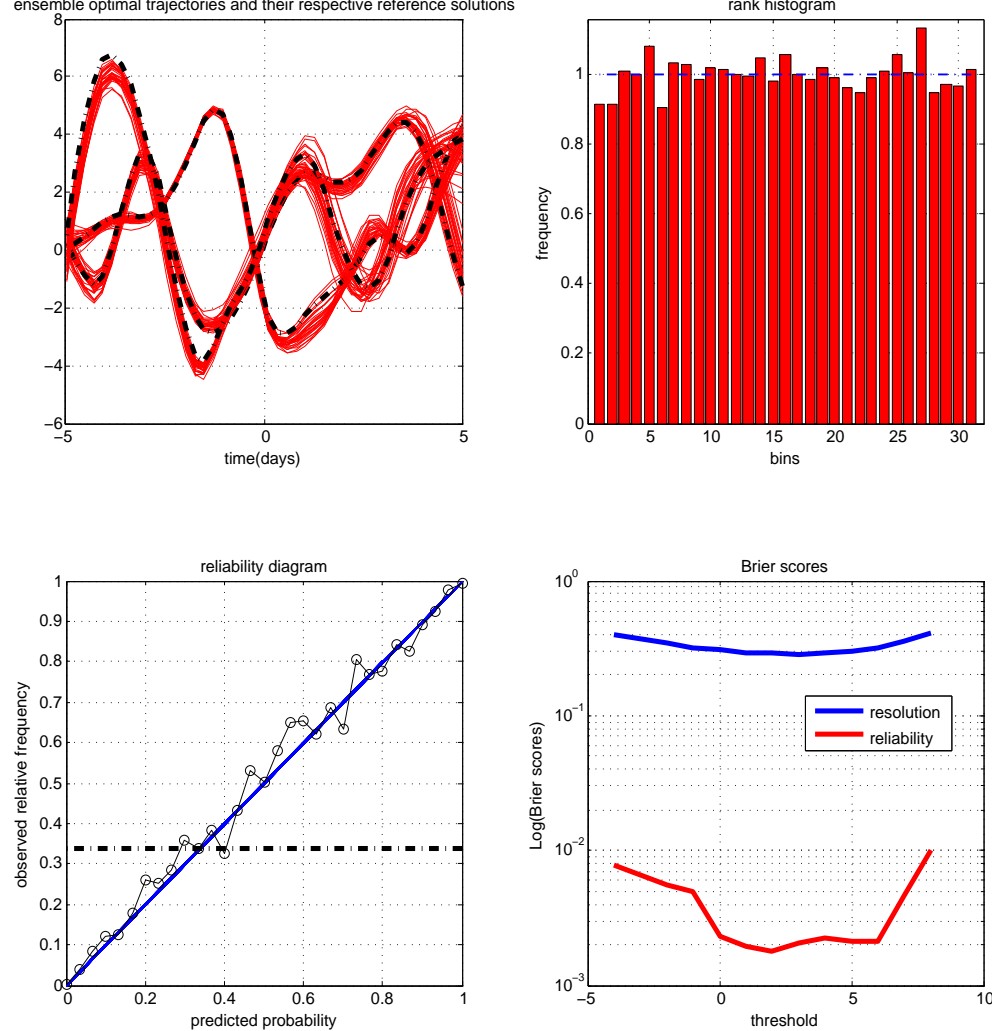

Fig. 12: Same as Figure 9, but at the end of 5-day forecasts. On the top-left panel the horizontal axis spans both the assimilation and the forecast intervals.



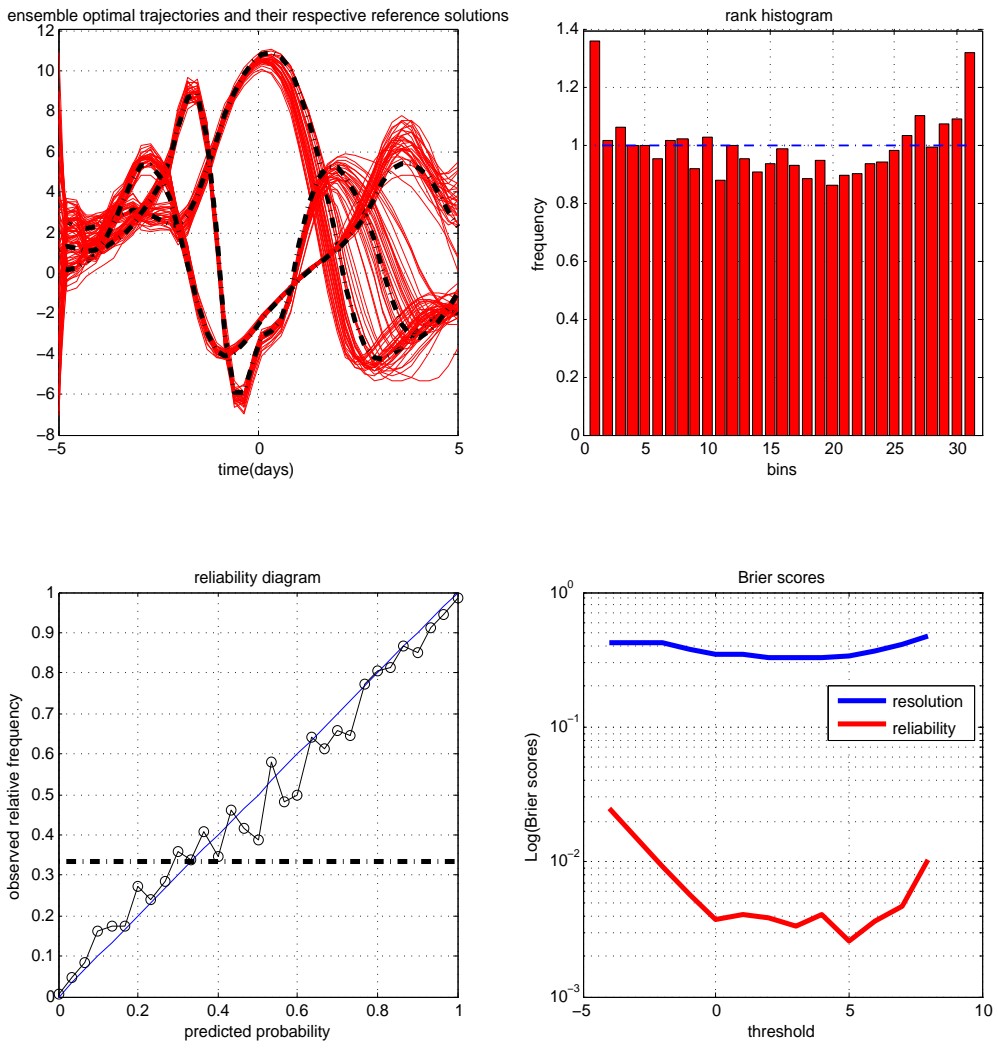

Fig. 13: Same as figure 12, but for EnKF.





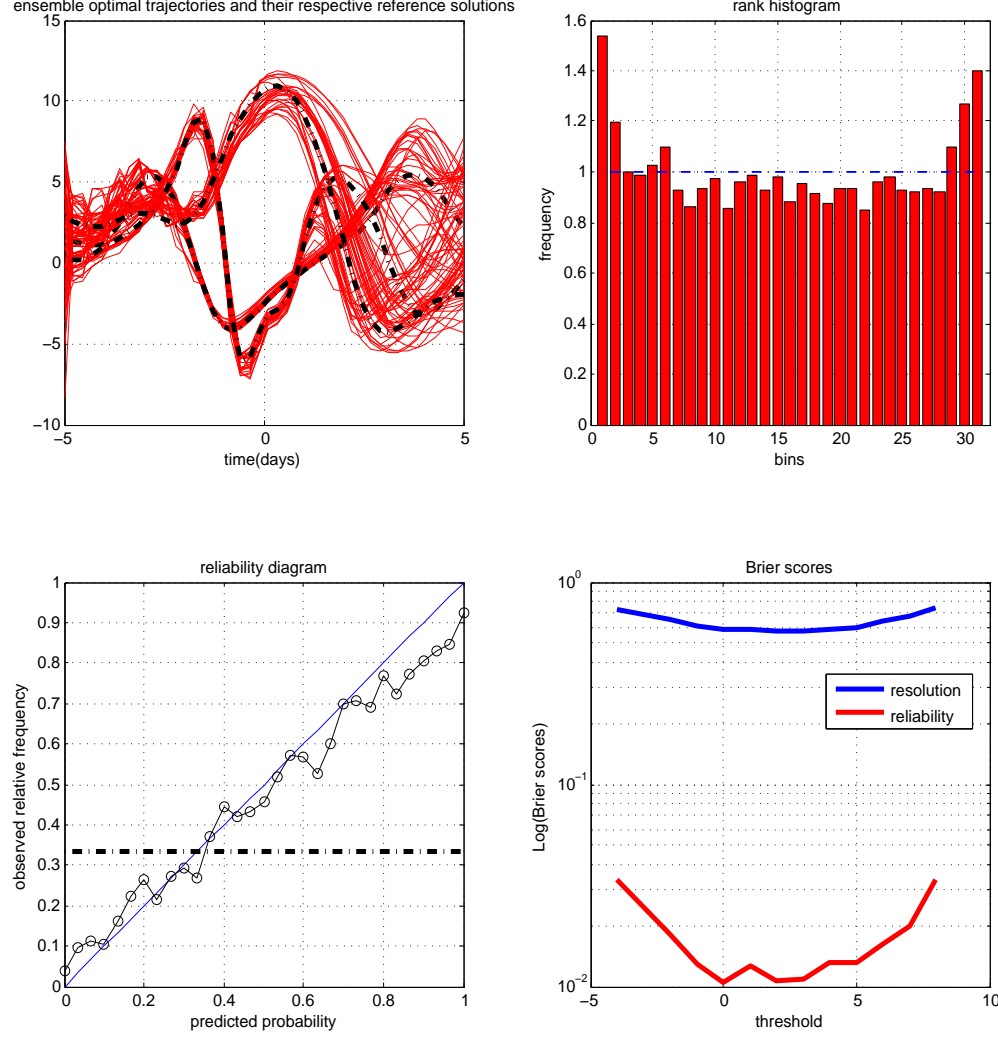

Fig. 14: Same as Figure 12, but for PF.





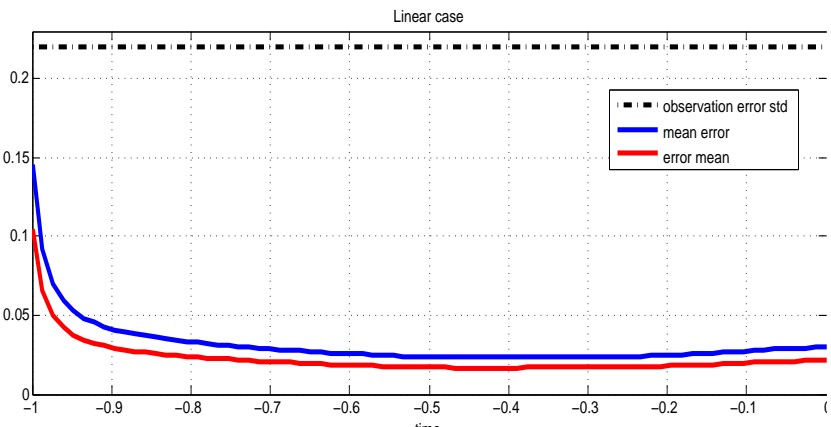

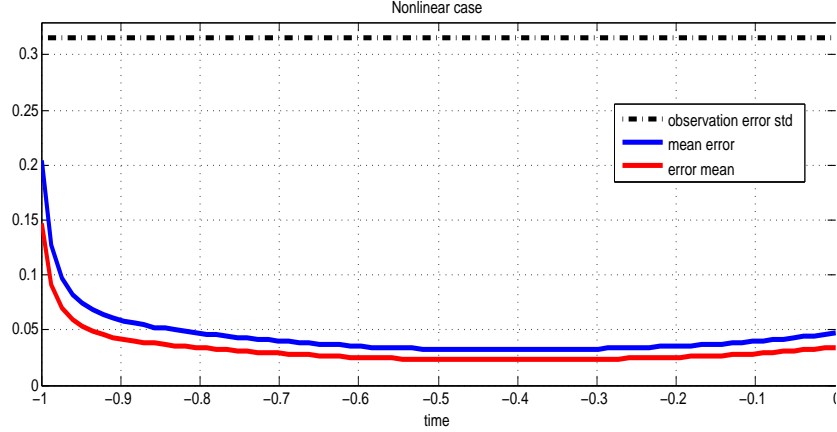

Fig. 15: Same as Figure 1, for variational ensemble assimilations performed on the Kuramoto-Sivashinsky equation, *i.e.* root-mean-square error from the truth along the assimilation window, averaged at each time over all grid points and all realizations, for both the linear and non-linear cases (top and bottom panels respectively).