# Peer review of "Ensemble Variational Assimilation as a Probabilistic Estimator. Part I: The linear and weak non-linear case"

_Nonlinear Processes in Geophysics, 2018_

## Referee Comment (RC1) · M. Bonavita (Referee) · 11 Feb 2018

The paper in subject is a valuable addition to the ongoing debate on the best strategy to perform ensemble data assimilation. The Authors evaluate the performance of an EDA type system (which they call EnsVAR) as a probabilistic estimator in linear and weakly-nonlinear regimes for two toy models. This is new, as most of the previous literature on the subject had focused on its performance as a deterministic estimator or as a tool to compute error estimates for a reference unperturbed analysis system. I found the paper interesting and well-written. Before recommending it for publication i suggests that some aspects, which I detail below, be further improved or better clarified.

[Figure]

1) Lines 54-56: I am not aware of the paper by le Gland et al, 2009, and I am not sure it is generally available as it appears to be an internal research memo of a specific institution. Further, I do not think it is actually necessary in this context, if we take the view that the EnKF converges to the KF for large ensemble size and the KF is a consistent bayesian estimator for linear dynamics and gaussian errors;

2) Lines 64-65: "They exist in numerous variants, many of which have been mathematically proven to achieve bayesianity in the limit of infinite ensemble size". Please provide relevant references;

3) Line 102-105: The Bardsley et al. 2014 reference appears to be missing. Some further discussion of their method would be useful here, as the response of the EnsVAR method to nonlinearities is the central issue of this paper;

4) Lines 177-179: I do not understand this remark and the implied derivations behind it. Can the Authors please expand?

5) Lines 4337-438: "We have evaluated the Gaussian character of the ensembles...by computing their negentropy". I suspect the Authors have verified the Gaussianity of some marginals of the full pdf, not the Gaussianity of the full multivariate distribution. Can the Authors be more specific on this pooint?

6) Lines 444-447: Can the lack of sensitivity of the analysis pdf to the pdf of the observations be considered a consequence of the Central Limit Theorem, or do the Authors have an alternative explanation?

7) I think it should be made clear that the comparison with the EnKF and PF is only qualitative, as the EnKF/PF results are known to be very sensitive to localization/inflation and there does not appear to have been a lot of work in this paper aimed at finding the optimal valuues;

8) Regarding the EnsVAR and EnKF comparison, I would expect the two systems to give equivalent results in the purely linear case. Have the Authors verified that this is

the case, or if it is not why?

---

## Referee Comment (RC2) · M. Bocquet (Referee) · 12 Mar 2018

I have been following the progress of M. Jardak and O. Talagrand on this work for about five years and I am very happy to finally see its fruits.

With a very few exceptions, it is a very well written and very enjoyable paper to read. The numerical experiments are carefully designed and defined (and averaged over sufficient runs for statistical significance), a quality often missing in similar papers. The discussion, theoretical and numerical tests are sophisticated and refined, and yet, all useful. To me, this clearly indicates that a lot of energy and intelligence have been put

into this study and paper.

There are a few flaws that need addressing, some of them requiring some care. However, they should be quick to address. Hence, I recommend a minor revision of the manuscript.

The weak points of the paper are:

1. An insufficient discussion in the introduction of the fact that we already know that naive RTO / EDA / EnsVAR is not (perfectly) Bayesian and proper references to Oliver et al. (1996); Bardsley et al. (2014); Liu et al. (2017) should be added or better referred to;

2. An abstract that – very surprisingly given how well the paper is written – does not faithfully reflect the findings of the paper;

3. Some technical issues such as:

   - The use of capital letters in the titles of the sections in an inconsistent manner; please check all titles.
   - Improper use of `\citet` and `\citep`. Please check this throughout the manuscript.
   - Many inconsistencies in how you refer to equations. This should be made consistent throughout the whole manuscript. Note that there is a recommended notation for Nonlinear Processes in Geophysics (see the guidelines).
   - One speaks of and writes "the ensemble Kalman filter", not "Ensemble Kalman Filter" (which could also let the reader think that you are not very familiar with the EnKF literature).
   - Spaces in between text and ":" need to be removed (especially in the figure captions).

Specific remarks, in connection, or not, to the previous remarks are:

1. Abstract: lines 5-7: even though the authors have worked on this idea for many years, the ensemble of data assimilation (EDA) principle implemented at Météo-France and ECMWF predates the EnsVAR and conceptually coincides with it. Moreover, it could be confused with the general terminology of EnVar (Ensemble Variational) methods. That is why I do not believe that this terminology should be put forward, especially in the abstract. That said, this obviously does not lessen the findings of the paper in any way. So, it is up to the authors.

2. Abstract, line 9: "the standard variational procedure" seems too vague. I am not sure of its meaning.

3. Abstract, line 10-14: the emphasis is on the performance (accuracy) of the method compared to, e.g., the EnKF. I do not believe that this is wise in the absence of a proper cycling with which the EnKF could shine. I do not understand why the emphasis is not on the discussion of the Bayesian (or not Bayesian) trait of the method and the quality of the updated ensemble, which is the strong point of this study.

4. Abstract, line 13: It is customary to write "the ensemble Kalman filter" instead of "Ensemble Kalman Filter".

5. lines 10-20: "of the observations proper": what do you mean?

6. line 41: The EnKF definitely uses a definite article and should read: "The ensemble Kalman filter...". This should be systematically checked throughout the manuscript.

7. line 42: "by of" $\longrightarrow$ "by".

8. line 80: "The present work is devoted to the study of that algorithm, and of its properties as a Bayesian estimator..." Precisely! That is why the abstract should reflect this point instead of focusing on the accuracy.

9. line 103-104: The connection between RTO and EDA as used in geophysical data assimilation has first been made, put forward and discussed in Liu et al. (2017) (and much more[1]). This must be mentioned here. (Incidentally this is how the authors of the present manuscript became aware of Bardsley et al. (2014).) Moreover, Oliver et al. and were the first to discuss this problem in 1996 (Oliver et al., 1996), which is something that Liu et al. (2017) recalled. You must cite this reference as well.

10. line 116: "succinctly": you could avoid the adverb, as it lets the reader think that it will not be enough information to get a clear idea on the numerical results.

11. line 124: "multi-dimensional/one-dimensional Gaussian" more precisely called "multivariate/univariate Gaussian".

12. line 125, before section 2: As already discussed (and illustrated) in Oliver et al. (1996); Bardsley et al. (2014); Liu et al. (2017), RTO/EDA (hence called naive RTO in Liu et al. (2017)) produces a biased nonlinear sampling. This should be briefly mentioned in the introduction as this is an established and published fact, important to your paper.

13. line 133: "data operator": why invent a new name when there exist "observation / forward / Jacobian / source-Receptor operator" in geophysical data assimilation?
* * *
[1](i) a shorter and heuristic derivation of Bardsley et al.' derivation, (ii) a higher dimensional illustration of RTO versus naive RTO, (iii) the suggestion and illustration that RTO is not as efficient as hoped for for higher-dimensional models, (iv) the claim that EDA as used in operational meteorology is (probably very moderately) impacted by this.

14. line 149: this statement only makes sense in an infinite numerical precision context. In practice, it of course depends on the condition number of operators built with $\Gamma$ and $\Sigma$.

15. line 175: "there is of course no reason to think...": again this has been settled in Oliver et al. (1996); Bardsley et al. (2014); Liu et al. (2017). So why not be more straightforward and factual here? Such as: "In general, this procedure does not lead to an unbiased Bayesian estimation, but can nonetheless provide a very useful approximation (Oliver et al., 1996; Bardsley et al., 2014; Liu et al., 2017)".

16. line 195: since it is here linear, you should use a bold upright character for H. See also line 291.

17. line 240-251: nice remark!

18. line 266: "bayesianity" $\longrightarrow$ "Bayesianity".

19. line 291: It is worth mentioning that this time-step is 0.05 time unit.

20. line 293: Since the results do not depend on $\sigma$, why not resort to the very commonly used (in data assimilation) value of $1$? Moreover, it is not the one used for the weakly nonlinear case. That may seem odd to the reader.

21. line 294: "(however, ...impact)": no need to place the statement within parentheses. In my opinion, your remark is legitimate and important.

22. lien 364: "in non-linear and non-Gaussian situations where Bayesianity does not hold" is not a consistent statement with your current introduction (you never mentioned this explicitly even when referring to Bardsley et al. (2014)). You should have referred to Oliver et al. (1996), Bardsley et al. (2014) (with the main result clearly mentioned) and Liu et al. (2017) for this statement to fully make sense.

23. line 369: "where one day is equal to 0.44 time unit...": I would rather say $0.20$ time unit(?).

24. line 375: "succesive" $\longrightarrow$ "successive".

25. line 392: "effect ." $\longrightarrow$ "effect.".

26. line 292: It is certainly a nonlinear effect, but do you have an explanation for it?

27. line 402: "(top, middle and bottom p and accuracy" should be removed.

28. lines 411-412: "The degradation of reliability in the lower two panels may therefore be due here to non-linearity.": this implies that you believe that naive RTO samples are not perfect draws from the conditional pdf. But, again, your introduction should have discussed this point more clearly.

29. lines 413-414: "is much larger for decrease of spatial density than for decrease of temporal density (middle and top panels respectively)." $\longrightarrow$ "is much larger for the decrease of spatial density than for the decrease of temporal density (middle and top panels respectively)."

30. line 416: "panelsrespectively" $\longrightarrow$ "panels respectively".

31. lines 415-416: "consistent with the top two panels of Figure 4, which suggest that the model fields are more correlated in time than in space...": yes, a very well known fact about the L96 model (with this configuration), which is actually what E. Lorenz wanted to achieve with this low-order model.

32. lines 440-442: You might want to slightly improve this discussion since comparing $10^{-3}$ to $10^{-6}$ could let the reader think the ensemble is actually very non-Gaussian (the appendix clarifies this point, but the text should not require the appendix to be clear).

33. line 449: "We present in this Section comparison..." $\longrightarrow$ "We present in this Section a comparison...".

34. lines 450-451: "Fair comparison is therefore possible only at the end of the assimilation window.": yes, but not only. A fair comparison of DA methods would also imply cycling, which is not the case of EnsVAR here. I am very fine about your using the EnKF and particle filter to compare the ensemble qualities; but not really when it comes to comparing RMSE at the end of the window. At the very least this should be briefly discussed.

35. line 459: "...is the one described by Evensen (2003)": which one? G. Evensen's book describes both stochastic and deterministic EnKFs. (Of course I know the answer, you just need to improve the statement.) By the way, you should, from time to time, insists on the fact that you picked up the stochastic EnKF since the deterministic is now more popular. Moreover, choosing the stochastic EnKF makes sense in this study as the EnsVAR is also stochastic. You could mention this as this would strengthen your choice for the stochastic EnKF.

36. line 461: With a fully observed system and an ensemble of size $N = 30$, you do not need to use localisation. In the present context it could actually be detrimental to the quality of the EnKF ensemble! (e.g., Bocquet and Carrassi, 2017). I would recommend that you do not use localisation here.

37. lines 479-481: It is fine to report these numbers in here, but not allude to them in the abstract, where, out of context, they do not make much sense.

38. line 503: "Kuramuto" $\longrightarrow$ "Kuramoto", as well as in both references by Yoshiki Kuramoto et al.

**References**

Bardsley, J.M., Solonen, A., Haario, H., Laine, M., 2014. Randomize-then-optimize: A method for sampling from posterior distributions in nonlinear inverse problems. SIAM J. Sci. Comput. 36, A1895–A1910.

Bocquet, M., Carrassi, A., 2017. Four-dimensional ensemble variational data assimilation and the unstable subspace. Tellus A 69, 1304504. doi:10.1080/16000870.2017.1304504.

Liu, Y., Haussaire, J.M., Bocquet, M., Roustan, Y., Saunier, O., Mathieu, A., 2017. Uncertainty quantification of pollutant source retrieval: comparison of bayesian methods with application to the Chernobyl and Fukushima-Daiichi accidental releases of radionuclides. Q. J. R. Meteorol. Soc. 143, 2886–2901. doi:10.1002/qj.3138.

Oliver, D.S., He, N., Reynolds, A.C., 1996. Conditioning permeability fields to pressure data, in: ECMOR V-5th European Conference on the Mathematics of Oil Recovery, pp. 259–269.

---

## Author Comment (AC1) · 9 Apr 2018

We thank M. Bonavita for his comments and suggestions. We give below a first response to some of these. Our responses are below the referee's comments and suggestions.

The paper in subject is a valuable addition to the ongoing debate on the best strategy to perform ensemble data assimilation. The Authors evaluate the performance of an EDA type system (which they call EnsVAR) as a probabilistic estimator in linear and weakly-nonlinear regimes for two toy models. This is new, as most of the previous literature on the subject had focused on its performance as a deterministic estimator or as a tool

to compute error estimates for a reference unperturbed analysis system. I found the paper interesting and well-written. Before recommending it for publication i suggests that some aspects, which I detail below, be further improved or better clarified.

1) Lines 54-56: I am not aware of the paper by le Gland et al, 2009, and I am not sure it is generally available as it appears to be an internal research memo of a specific institution. Further, I do not think it is actually necessary in this context, if we take the view that the EnKF converges to the KF for large ensemble size and the KF is a consistent bayesian estimator for linear dynamics and gaussian errors;

We think the reference is appropriate since it is certainly useful to know for sure that EnKF cannot be Bayesian in general in the nonlinear case.

3) Line 102-105: The Bardsley et al. 2014 reference appears to be missing. Some further discussion of their method would be useful here, as the response of the EnsVAR method to nonlinearities is the central issue of this paper;

The Bardsley et al. paper is actually referenced in the maunscript, but at the wrong alphabetical place. And the question of further discussion has also been raised with some emphasis by Reviewer 2. We will modify our paper accordingly.

4) Lines 177-179: I do not understand this remark and the implied derivations behind it. Can the Authors please expand?

Since Bayesianity is ensured under the two conditions of linearity and Gaussianity, one can legitimately wonder what can be said when only linearity is present. That is the reason for our remark. We will reformulate it in order to make our point clearer.

6) Lines 444-447: Can the lack of sensitivity of the analysis pdf to the pdf of the observations be considered a consequence of the Central Limit Theorem, or do the Authors have an alternative explanation?

The fact, which has been mentioned just before, that the ensembles produced by the assimilation are close to Gaussianity is certainly a consequence of the Central Limit

Theorem. That the ensembles are insensitive to the shape of the pdf of the errors in the observations seems rather natural in these conditions, but we cannot really say more at this stage.

8) Regarding the EnsVAR and EnKF comparison, I would expect the two systems to give equivalent results in the purely linear case. Have the Authors verified that this is the case, or if it is not why?

Yes, one could expect similar results in the linear (and also Gaussian) case from EnsVAR and EnKF, since both are then bayesian (as indeed PF is also supposed to be). But there is actually a slight difference. EnsVAR produces a set of independent realizations of the conditional pdf for any ensemble size, while EnsVAR or PF do not. And we have not made the comparison. The reason for that is that our paper is concentrated on EnsVAR, and we have used the linear and Gaussian case, not for evaluating it per se, but as a benchmark for evaluation of the nonlinear case.
* * *

---

## Author Comment (AC2) · 9 Apr 2018

We thank M. Bocquet for his comments and suggestions. We give below a first response to some of these. The referee's comments and suggestions are on top of our responses.

The weak points of the paper are:

1. An insufficient discussion in the introduction of the fact that we already know that naive RTO / EDA / EnsVAR is not (perfectly) Bayesian and proper references to Oliver et al. (1996); Bardsley et al. (2014); Liu et al. (2017) should be added or better referred

to.

Thanks. We will mention these papers and discuss them in the perspective of our two papers.

2. An abstract that – very surprisingly given how well the paper is written – does not faithfully reflect the findings of the paper.

We do not fully understand what the referee means. But see our response below to the reviewer's specific remark 3.

Specific remarks, in connection, or not, to the previous remarks are:

3. Abstract, line 10-14: the emphasis is on the performance (accuracy) of the method compared to, e.g., the EnKF. I do not believe that this is wise in the absence of a proper cycling with which the EnKF could shine. I do not understand why the emphasis is not on the discussion of the Bayesian (or not Bayesian) trait of the method and the quality of the updated ensemble, which is the strong point of this study.

By writing performance, we did not mean specifically numerical accuracy. We meant global performance of the algorithms under comparison, and primarily their performance as Bayesian estimators. We will modify the wording so as to avoid any misunderstanding. As for cycling of EnsVAR, that is certainly an important and interesting point, but we do not think it is directly relevant to our paper, since we have compared the three algorithms (EnsVAR, EnKF and PF) over the same assimilation windows.

8. line 80: "The present work is devoted to the study of that algorithm, and of its properties as a Bayesian estimator..." Precisely! That is why the abstract should reflect this point instead of focusing on the accuracy.

See previous point 3.

9. line 103-104: The connection between RTO and EDA as used in geophysical data assimilation has first been made, put forward and discussed in Liu et al. (2017) (and

much more1). This must be mentioned here. (Incidentally this is how the authors of the present manuscript became aware of Bardsley et al. (2014).) Moreover, Oliver et al. and were the first to discuss this problem in 1996 (Oliver et al., 1996), which is something that Liu et al. (2017) recalled. You must cite this reference as well.

See 'weak point' 1 above.

12. line 125, before section 2: As already discussed (and illustrated) in Oliver et al. (1996); Bardsley et al. (2014); Liu et al. (2017), RTO/EDA (hence called naïve RTO in Liu et al. (2017)) produces a biased nonlinear sampling. This should be briefly mentioned in the introduction as this is an established and published fact, important to your paper.

Again, see 'weak point' 1 above.

13. line 133: "data operator": why invent a new name when there exist "observation /forward / Jacobian / source-Receptor operator" in geophysical data assimilation?

We want to stress here (see ll. 136-140) that the data vector z contains all the information to be used for estimating the state vector x (physical observations, complete or partial background(s) or a priori estimates, 'balance' conditions or dynamical equations to be verified to some degree of accuracy by the final estimate, . . .). The expressions suggested by the referee are more restrictive than that. We will stress more strongly what we mean.

15. line 175: "there is of course no reason to think...": again this has been settled in Oliver et al. (1996); Bardsley et al. (2014); Liu et al. (2017). So why not be more straightforward and factual here? Such as: "In general, this procedure does not lead to an unbiased Bayesian estimation, but can nonetheless provide a very useful approximation (Oliver et al., 1996; Bardsley et al., 2014; Liu et al., 2017)".

Again, see 'weak point' 1 above.

34. lines 450-451: "Fair comparison is therefore possible only at the end of the assimilation window.": yes, but not only. A fair comparison of DA methods would also imply cycling, which is not the case of EnsVAR here. I am very fine about your using the EnKF and particle filter to compare the ensemble qualities; but not really when it comes to comparing RMSE at the end of the window. At the very least this should be briefly discussed.

We do not really understand what the referee means here. What we mean is that it is only at the end of the assimilation window that the three algorithms have used the same amount of information, and that it is only at that time that comparison is fair, in terms of Bayesianity as well as RMSE. Having a form of cycling for EnsVAR within the overall assimilation window would define another algorithm, which could also be compared to what we have obtained. But we do not understand in what that would be 'fairer'.

35. line 459: "...is the one described by Evensen (2003)": which one? G. Evensen's book describes both stochastic and deterministic EnKFs. (Of course I know the answer, you just need to improve the statement.) By the way, you should, from time to time, insists on the fact that you picked up the stochastic EnKF since the deterministic is now more popular. Moreover, choosing the stochastic EnKF makes sense in this study as the EnsVAR is also stochastic. You could mention this as this would strengthen your choice for the stochastic EnKF.

We compare three algorithms which use the same input (data and a priori statistical information about the quality of those data). Whether or not these algorithms are stochastic (in the sense that they use random generators at some stage, if that is what the referee means) is secondary in our mind.

36. line 461: With a fully observed system and an ensemble of size N = 30, you do not need to use localisation. In the present context it could actually be detrimental to the quality of the EnKF ensemble! (e.g., Bocquet and Carrassi, 2017). I would recommend that you do not use localisation here.

A number of other parameters than localisation could be evaluated there spatio-temporal distribution of the observations, statistical properties of the observation errors, . . .). Our purpose is not to make a thorough study of the many variants that could be implemented for EnKF of PF. It is just to compare EnsVAR with what seems to us to be two 'reasonable' algorithms for both EnKF and PF. And we say clearly that our conclusions on the comparison of the three algorithms cannot be considered as definitive.

37. lines 479-481: It is fine to report these numbers in here, but not allude to them in the abstract, where, out of context, they do not make much sense.

Actually, it was not our intention to allude to these numbers in our abstract (see our response to specific remark 3 above).

---

## Author Response (AR1)

**Answers to referees: npg-2018-5, 2018**

**Mohamed Jardak & Olivier Talagrand**

**1   To referee 1:**

We thank M. Bonavita for his comments and suggestions. These are printed below in black, and our responses in red.

1. Lines 54-56: I am not aware of the paper by le Gland et al, 2009, and I am not sure it is generally available as it appears to be an internal research memo of a specific institution. Further, I do not think it is actually necessary in this context, if we take the view that the EnKF converges to the KF for large ensemble size and the KF is a consistent bayesian estimator for linear dynamics and gaussian errors. We have given a different, more accessible, reference. And, since we are concerned with the Bayesianity of ensemble estimates, we think it is legitimate to consider the Bayesianity of the EnKF, and in particular to stress that it cannot be Bayesian in the general nonlinear case.

2. Lines 64-65: "They exist in numerous variants, many of which have been mathematically proven to achieve bayesianity in the limit of infinite ensemble size". Please provide relevant references.
   We have added a reference which deals with the general question of asymptotic convergence of Particle Filters.

3. Line 102-105: The Bardsley et al. 2014 reference appears to be missing. Some further discussion of their method would be useful here, as the response of the EnsVAR method to nonlinearities is the central issue of this paper. The Bardsley et al. paper was actually referenced in the manuscript, but at the wrong alphabetical place. And the question of further discussion has also been raised with some emphasis by Referee 2. See his review, especially his 'weak point' 1, and our response to it.

4. Lines 177-179: I do not understand this remark and the implied derivations behind it. Can the Authors please expand? Since Bayesianity is ensured in the conditions of linearity and Gaussianity, it is legitimate to consider the case when linearity holds, but not Gaussianity. As for "the implied derivations", if the referee means how the results we state can be obtained, that is actually trivial (and we have given a reference anyway).

5. Lines 4337-438: "We have evaluated the Gaussian character of the ensembles...by computing their negentropy". I suspect the Authors have verified the Gaussianity of some marginals of the full pdf, not the Gaussianity of the full multivariate distribution. Can the Authors be more specific on this point? Correct, we never evaluated the Gaussianity for the full multivariate distribution. We have now made that clearer.

6. Lines 444-447: Can the lack of sensitivity of the analysis pdf to the pdf of the obser- vations be considered a consequence of the Central Limit Theorem, or do the Authors have an alternative explanation?
The fact, which has been mentioned just before, that the ensembles produced by the assimilation are close to Gaussianity is certainly a consequence of the Central Limit Theorem. That the ensembles are insensitive to the shape of the pdf of the errors in the observations seems rather natural in these conditions, but we cannot really say more at this stage.

7. I think it should be made clear that the comparison with the EnKF and PF is only qualitative, as the EnKF/PF results are known to be very sensitive to localiza- tion/inflation and there does not appear to have been a lot of work in this paper aimed at finding the optimal values. We give numerical results, and the comparison is quantitave. But it is certainly not exhaustive. We had alredy stressed that point in the paper, which has also been raised by the other referee.

8. Regarding the EnsVAR and EnKF comparison, I would expect the two systems to give equivalent results in the purely linear case. Have the Authors verified that this is the case, or if it is not why?
Yes, one could expect similar results in the linear (and also Gaussian) case from EnsVAR and EnKF, since both are then bayesian (as so is PF). But there is actually a slight difference. EnsVAR produces a set of independent realizations of the conditional pdf for any ensemble size, while EnsVAR or PF do not. We have not made the comparison. The reason for that is that our paper is concentrated on EnsVAR, and we have used the linear and Gaussian case, not for evaluating it per se, but as a benchmark for evaluation of the nonlinear case

**2   To referee 2 :**

We thank M. Bocquet for his comments and suggestions, *particularly for the references he has mentioned* . These are printed below in black, and our responses in red.

1. An insufficient discussion in the introduction of the fact that we already know that naive RTO / EDA / EnsVAR is not (perfectly) Bayesian and proper references to Oliver et al. (1996); Bardsley et al. (2014); Liu et al. (2017) should be added or better referred to.
Thanks. We have mentioned these papers and discuss them in the perspective of our two papers.

2. An abstract that – very surprisingly given how well the paper is written – does not faithfully reflect the findings of the paper. We do not fully understand what the referee means. But see our response below to his specific remark 3.

3. Some technical issues such as:

   - The use of capital letters in the titles of the sections in an inconsistent manner; please check all titles.
   - Improper use of *citet* and *citep* Please check this throughout the manuscript.
   - Many inconsistencies in how you refer to equations. This should be made consistent throughout the whole manuscript. Note that there is a recommended notation for Nonlinear Processes in Geophysics (see the guide- lines).
   - One speaks of and writes "the ensemble Kalman filter", not "Ensemble Kalman Filter" (which could also let the reader think that you are not very familiar with the EnKF literature).
   - Spaces in between text and ":" need to be removed (especially in the figure captions).

   Thanks, the above points have been fixed

1. Abstract: lines 5-7: even though the authors have worked on this idea for many years, the ensemble of data assimilation (EDA) principle implemented at Météo-France and ECMWF predates the EnsVAR and conceptually coincides with it. Moreover, it could be confused with the general terminology of EnVar (Ensemble Variational) methods. That is why I do not believe that this terminology should be put forward, especially in the abstract. That said, this obviously does not lessen the findings of the paper in any way. So, it is up to the authors. Thank you, we have stated explicitly in the abstracts of both parts of the paper, and in the texts,that EnsVAR is the same thing as EDA

2. Abstract, line 9: "the standard variational procedure" seems too vague. I am not sure of its meaning. We meant variational assimilation, as described and used in numerous papers and in operational prediction in places like ECMWF and Météo-France. We do not think that is too vague. We do not want to overload the abstract.

3. Abstract, line 10-14: the emphasis is on the performance (accuracy) of the method compared to, e.g., the EnKF. I do not believe that this is wise in the ab- sence of a proper cycling with which the EnKF could shine. I do not understand why the emphasis is not on the discussion of the Bayesian (or not Bayesian) trait of the method and the quality of the updated ensemble, which is the strong point of this study. By writing performance, we did not mean specifically numerical accuracy. We meant global performance of the algorithms under comparison, and primarily their performance as Bayesian estimators. We have modified the wording so as to avoid any misunderstanding.

4. Abstract, line 13: It is customary to write "the ensemble Kalman filter" instead of "Ensemble Kalman Filter". Thanks, fixed.

5. lines 10-20: "of the observations proper": what do you mean? Thanks, we meant physical observations. We have modified the wording so as to avoid any misunderstanding.

6. line 41: The EnKF definitely uses a definite article and should read: "The ensemble Kalman filter...". This should be systematically checked throughout the manuscript. This is the same comment as comment 4 above. Thanks, fixed

7. line 42: "by of" $\rightarrow$ "by". Thanks, fixed

8. line 80: "The present work is devoted to the study of that algorithm, and of its properties as a Bayesian estimator..." Precisely! That is why the abstract should reflect this point instead of focusing on the accuracy. See comment 3 above, and our response to it.

9. line 103-104: The connection between RTO and EDA as used in geophysical data assimilation has first been made, put forward and discussed in Liu et al. (2017) (and much more1). This must be mentioned here. (Incidentally this is how the authors of the present manuscript became aware of Bardsley et al. (2014).) Moreover, Oliver et al. and were the first to discuss this problem in 1996 (Oliver et al., 1996), which is something that Liu et al. (2017) recalled. You must cite this reference as well. We have added all these references, and commented on their connection with the present work.

10. line 116: "succinctly": you could avoid the adverb, as it lets the reader think that it will not be enough information to get a clear idea on the numerical results. Thanks, done

11. line 124: "multi-dimensional/one-dimensional Gaussian" more precisely called "multivariate/univariate Gaussian". Thanks, done.

12. line 125, before section 2: As already discussed (and illustrated) in Oliver et al. (1996); Bardsley et al. (2014); Liu et al. (2017), RTO/EDA (hence called naive RTO in Liu et al. (2017)) produces a biased nonlinear sampling. This should be briefly mentioned in the introduction as this is an established and published fact, important to your paper. Thanks, references added and commented on as it can be seen in the introduction.

13. line 133: "data operator": why invent a new name when there exist "observation / forward / Jacobian / source-Receptor operator" in geophysical data assimilation? We want to stress here (see ll. 136-140) that the data vector $\mathbf{z}$ contains all the information to be used for estimating the state vector $\mathbf{x}$ (physical observations, complete or partial background(s) or a priori estimates, 'balance' conditions or dynamical equations to be verified to some degree of accuracy by the final estimate,

...). The expressions suggested by the referee are usually more restrictive than that. And the expression "data operator" has indeed been used in the literature.

14. line 149: this statement only makes sense in an infinite numerical precision context. In practice, it of course depends on the condition number of operators built with $\Gamma$ and $\Sigma$. Yes, of course. But strict mathematical results are always useful to start with, before considering numerical aspects.

15. line 175: "there is of course no reason to think...": again this has been settled in Oliver et al. (1996); Bardsley et al. (2014); Liu et al. (2017). So why not be more straightforward and factual here? Such as: "In general, this procedure does not lead to an unbiased Bayesian estimation, but can nonetheless provide a very useful approximation (Oliver et al., 1996; Bardsley et al., 2014; Liu et al., 2017)". All right. Corrected.

16. line 195: since it is here linear, you should use a bold upright character for H. See also line 291. Thanks, fixed.

17. line 240-251: nice remark! Thanks.

18. line 266: "bayesianity" $\rightarrow$ "Bayesianity". Thanks, fixed.

19. line 291: It is worth mentioning that this time-step is 0.05 time unit. We have introduced the "day" as equal to 0.44 time unit in equation 12.

20. line 293: Since the results do not depend on $\sigma$ , why not resort to the very commonly used (in data assimilation) value of 1 ? Moreover, it is not the one used for the weakly nonlinear case. That may seem odd to the reader. Even though the results are independent of the value $\sigma$, one value must be used for the numerical computations. We mention the value we have used. And we write now in Section 5, relative to the nonlinear case, that it is necessary to scale the value of $\sigma$ with the varaibility of the model.

21. line 294: "(however, ...impact)": no need to place the statement within parentheses. In my opinion, your remark is legitimate and important. Thanks, parentheses removed.

22. line 364: "in non-linear and non-Gaussian situations where Bayesianity does not hold" is not a consistent statement with your current introduction (you never mentioned this explicitly even when referring to Bardsley et al. (2014)). You should have referred to Oliver et al. (1996), Bardsley et al. (2014) (with the main result clearly mentioned) and Liu et al. (2017) for this statement to fully make sense. All right, corrected.

23. line 369: "where one day is equal to 0.44 time unit...": I would rather say 0.20 time unit(?). That is actually our definition of a 'day'.

24. line 375: "succesive" $\rightarrow$ "successive". Thanks fixed.

25. line 392: "effect ." → "effect.". Thanks fixed.

26. line 392: It is certainly a nonlinear effect, but do you have an explanation for it? No, we do not have a clear explanation for it. We nevertheless mention now that, since the conditional expectation is the deterministic estimator which minimises the variance of the estimation error, it must lead in general to a smaller error variance than the deterministic assimilation performed on the raw observations.

27. line 402: "(top, middle and bottom p and accuracy" should be removed. Thanks, fixed.

28. lines 411-412: "The degradation of reliability in the lower two panels may therefore be due here to non-linearity.": this implies that you believe that naive RTO samples are not perfect draws from the conditional pdf. But, again, your introduc- tion should have discussed this point more clearly. As already said, this is now done.

29. lines 413-414: "is much larger for decrease of spatial density than for decrease of temporal density (middle and top panels respectively)." → "is much larger for the decrease of spatial density than for the decrease of temporal density (middle and top panels respectively)." We are not convinced, and leave that to the text editor.

30. line 416: "panelsrespectively" *rightarrow* "panels respectively". Thanks, fixed

31. lines 415-416: "consistent with the top two panels of Figure 4, which suggest that the model fields are more correlated in time than in space...": yes, a very well known fact about the L96 model (with this configuration), which is actually what E. Lorenz wanted to achieve with this low-order model. Thank you. We didn't know that this point is a well known fact.

32. . lines 440-442: You might want to slightly improve this discussion since compar- ing $10^-3$ to $10^-6$ could let the reader think the ensemble is actually very non-Gaussian (the appendix clarifies this point, but the text should not require the appendix to be clear). Thank, your comment has been taken into account and the text has been modified for more clarity.

33. line 449: "We present in this Section comparison..." → "We present in this Section a comparison...". Thanks, done.

34. lines 450-451: "Fair comparison is therefore possible only at the end of the as- similation window.": yes, but not only. A fair comparison of DA methods would also imply cycling, which is not the case of EnsVAR here. I am very fine about your using the EnKF and particle filter to compare the ensemble qualities; but not really when it comes to comparing RMSE at the end of the window. At the very least this should be briefly discussed. We do not really understand what the referee means here. What we mean is that it is only at the end of the assimilation window that the three algorithms have used the same amount of information, and that it is only at that time that comparison is fair, in terms of Bayesianity as well

as RMSE. Having a form of cycling for EnsVAR within the overall assimilation window would define another algorithm, which could also be compared to what we have obtained. But we do not understand in what that would be 'fairer'.

35. line 459: "...is the one described by Evensen (2003)": which one? G. Evensen's book describes both stochastic and deterministic EnKFs. (Of course I know the answer, you just need to improve the statement.) By the way, you should, from time to time, insists on the fact that you picked up the stochastic EnKF since the deterministic is now more popular. Moreover, choosing the stochastic EnKF makes sense in this study as the EnsVAR is also stochastic. You could mention this as this would strengthen your choice for the stochastic EnKF. Yes, we have used a stochastic EnKF. This has now been added in the paper. Now, we do not see any necessary connection between the stochastic character of our two algorithms, if 'stochastic' only means that the data are perturbed at some stage in the algorithm.

36. line 461: With a fully observed system and an ensemble of size $N = 30$ , you do not need to use localisation. In the present context it could actually be detrimental to the quality of the EnKF ensemble! (e.g., Bocquet and Carrassi, 2017). I would recommend that you do not use localisation here. Following comments from both referees, we have made a few experiments not using localisation in the EnKF. The RMSE and the RCRV are significantly degraded, while the rank histogram and the resolution component of the Brier score are improved. The reliability component of the Brier score remained the same. All that is true for both assimilation and forecast. These results, only mentioned but not actually presented in the paper, would deserve further studies which are postponed for a future work.

37. lines 479-481: It is fine to report these numbers in here, but not allude to them in the abstract, where, out of context, they do not make much sense. Actually, it was not our intention to allude to these numbers in our abstract (see our response to point 6).

38. line 503: "Kuramuto" → "Kuramoto", as well as in both references by Yoshiki Kuramoto et al. Thanks, fixed.

---

## Referee Report (RR1)

Review of *Ensemble Variational Assimilation as a Probabilistic Estimator. Parts I and II (revision)* by M. Jardak and O. Talagrand Sumitted to Nonlin. Processes Geophys. A. Carrassi Editor.

Marc Bocquet

July 18, 2018

I thank the authors for their answers and their revision. I recommend acceptance of both manuscripts. There are a few points about which the authors might want to give it a second thought. In the following, I am not commenting on the choices made by the authors, even when I would have done otherwise. I only comment when I believe they did not get something significant. If need be, any technical correction that the authors would like to make can be made prior to uploading the final draft, under the control of the Editor.

**1 Ensemble Variational Assimilation as a Probabilistic Estimator. Part I**

- 3. Abstract, line 10-14: the emphasis is on the performance (accuracy) of the method compared to, e.g., the EnKF. I do not believe that this is wise in the absence of a proper cycling with which the EnKF could shine. I do not understand why the emphasis is not on the discussion of the Bayesian (or not Bayesian) trait of the method and the quality of the updated ensemble, which is the strong point of this study. By writing performance, we did not mean specifically numerical accuracy. We meant global performance of the algorithms under comparison, and primarily their performance as Bayesian estimators. We have modified the wording so as to avoid any misunderstanding.

  You did not really answer to this point. You now mention both Bayesian estimator and deterministic estimator. Hence, my remark is still valid. In the absence of cycling of the EnsVar, you have not shown that the EnsVar outperforms the EnKF, at least as a deterministic estimator. To be clear: the comparison that you make in the abstract is the outcome of your experiments. But these experiments do not reflect the general goal of data assimilation for the geofluids, which require cycling over a long period of time. I think this is totally acceptable for the Bayesian estimator, but not for the deterministic estimator. There is nothing wrong with your results but you should contextualise them and mention their limitation.

- 9. line 103-104: The connection between RTO and EDA as used in geophysical data assimilation has first been made, put forward and discussed in Liu et al. (2017) (and much more1). This must be mentioned here. (Incidentally this is how the authors of the present manuscript became aware of Bardsley et al. (2014).) More- over, Oliver et al. and were the first to discuss this problem in 1996 (Oliver et al., 1996), which is something that Liu et al. (2017) recalled. You must cite this reference as well. We have added all these references, and commented on their connection with the present work.

  You should mention the name RTO here too, since it is used by several authors.

- 19. line 291: It is worth mentioning that this time-step is 0.05 time unit. We have introduced the "day" as equal to 0.44 time unit in equation 12.

  My remark is still valid. Your presentation is unclear and unconventional. The reader has no way to know at this stage what this time-step is.

- line 369: "where one day is equal to 0.44 time unit...": I would rather say 0.20 time unit(?). That is actually our definition of a 'day'.

  This is not Lorenz's definition of a day for his model, and not the definition used by all of your colleagues. This is very confusing.

- lines 450-451: "Fair comparison is therefore possible only at the end of the assimilation window.": yes, but not only. A fair comparison of DA methods would also imply cycling, which is not the case of EnsVAR here. I am very fine about your using the EnKF and particle filter to compare the ensemble qualities; but not really when it comes to comparing RMSE at the end of the window. At the very least this should be briefly discussed. We do not really understand what the referee means here. What we mean is that it is only at the end of the assimilation window that the three algorithms have used the same amount of information, and that it is only at that time that comparison is fair, in terms of Bayesianity as well as RMSE. Having a form of cycling for EnsVAR within the overall assimilation window would define another algorithm, which could also be compared to what we have obtained. But we do not understand in what that would be 'fairer'.

  You have to contextualise your results. Your comparison between the EnsVar and the EnKF is established in the restricted context of your numerical experiments. It turns out that your experiments are not that general because you do not cycle EnsVar. Hence you cannot make general claim about the EnKF. A the very least, you have to recall the absence of cycling of the EnsVar. Again, the primary goal of data assimilation in meteorology/oceanography is the long-term tracking of the state of a geofluid. You have not defined an algorithm suited for this (since it cannot extend beyond a certain time horizon). I am fine with your choice. But you have to discuss this and contextualise.

- 35. line 459: "...is the one described by Evensen (2003)": which one? G. Evensen's book describes both stochastic and deterministic EnKFs. (Of course I know the answer, you just need to improve the statement.) By the way, you should, from time to time, insists on the fact that you picked up the stochastic EnKF since the deterministic is now more popular. Moreover, choosing the stochastic EnKF makes sense in this study as the EnsVAR is also stochastic. You could mention this as this would strengthen your choice for the stochastic EnKF. Yes, we have used a stochastic EnKF. This has now been added in the paper. Now, we do not see any necessary connection between the stochastic character of our two algorithms, if 'stochastic' only means that the data are perturbed at some stage in the algorithm.

  Yes, 'stochastic' only means that the data are perturbed at some stage in the algorithm. This algorithmic connection is very strong, and very well know. Try comparing with a deterministic EnKF: you will understand then!

**2 Ensemble Variational Assimilation as a Probabilistic Estimator. Part II**

- lines 24-25: "The performance of EnsVAR is compared with that of Ensemble Kalman Filter and Particle Filter in Section 3.": again, out of a specific context, this does not make much sense in the absence of cycling, proper tuning of the methods, and so on. We have mentioned (ll. 305-310) that the comparison with EnKF and PF certainly cannot be considered as definitely conclusive. But it is certainly instructive, for instance in that it suggests that there are no major differences between the results produced by the three methods that have been compared. And we do not understand why the referee considers that 'this does not make much sense in the absence of cycling' (see our response to his specific remark 34 about paper 1). And 'proper tuning of the methods' could be an endless task.

  Please see my response to the first paper authors' response. It is the sentence "and its performance is as least as good as" that causes trouble because it is not put into the rather narrow context of the experiments of the paper. No, 'proper tuning of the methods' is not an endless task (but certainly tedious), otherwise we would not be bench-marking EnKF methods. This is usually a requirement.

- line 28: "successful in nonlinear as in linear conditions.": it always depends on how long the data assimilation window is. As any other method, EnsVAR is bound to fail for very large windows. We qualified our statement by saying that it is valid only for the conditions of our experiments. But is not clear to us why any method is bound to fail for very large windows. Failure is certainly to be expected for strong constraint assimilation implemented with an erroneous model. But why should it be in the case of weak constraint ?

  As opposed to the EnKF, nobody has ever been able to set up a (single analysis) 4D-Var for chaotic models over very long windows. For the strong constraint 4D-Var, it is due to the dynamical

instability of the model. For the weak-constraint 4D-Var, this is due to the numerical cost (storage, computational). Even for stable dynamics (atmospheric chemistry for instance), it is customary to split the 4D-Var analyses over several segments of a few weeks each.

- 7. line 33: "twice a day": please mention that this corresponds to 0.10 time units since the Lorenz model is primarily defined in those units. We have defined the "day" in paper I as equal to 0.44 time unit in equation I.12.

  The day has been defined by Lorenz as 0.20 time unit. Most colleagues stick to this (not so arbitrary actually) definition. Your choice may generate a lot of confusion.

---

## Author Response (AR2)

**Answers to referees and editor : npg-2018-5 &6, 2018**

Mohamed Jardak & Olivier Talagrand

**1 To referee I:**

We thank the referee for his suggestions for future research, concerning in particular the numerical cost of EnsVAR.

**2 To referee II:**

1. The referee has spotted an inconsistency in our paper, for which we thank him. We have made the correction. Our 'day' is equal to 0.24 time unit in Equation (12) (instead of 0.2 in the paper by Lorenz). We do not think the difference is critical.

2. As requested by the referee, we have explicitly mentioned the Random-Then-Optimize (RTO ) algorithm in our Introduction (it was already mentioned in our Conclusion).

3. The other comments of the referee bear on what he considers are limitations of our work and of our conclusions. We basically agree with him, and we had already mentioned that our conclusions are limited to the conditions of our experiments. The referee mentions deterministic versions EnKF as an alternative to the stochastic version we have used. We now include the use of deterministic EnKF among the various possibilities for future works. The referee stresses that our EnsVAR is not cycled, and seems to consider that, because of the ensuing numerical cost, it could not be used in practical situations. That may the case, and cycling is already discussed in our papers, in particular in the perspective of future works. In the other hand, we do not understand some of the remarks made by the referee on this aspect of cycling. He writes for instance I think this is totally acceptable for the Bayesian estimator, but not for the deterministic estimator (where this designates our approach). We do not understand why the referee makes here a difference between the two estimators. In our logic, assimilation is intrinsically a problem in Bayesian estimation, and a deterministic estimator can only be a by-product (e.g., an expectation) of a Bayesian estimator. In any case, the fact that our EnsVAR is not cycled is stressed in two places in our Part I, and discussed again in the Conclusion of Part II. We do not think it is necessary to add more on this aspect.

**3 To the editor:**

The Editor has specifically asked us to consider two points. One is the question of the time unit we use. We think this has now been clarified. Our 'day' is equal to 0.24 time unit in Equation (12) (see point 1 in our response to Referee 2). As for the other point, the editor writes I am also concerned about the lack of a clear assertion that your experiments do not include any cycling. Well, the fact that our experiments do not include any cycling was clearly asserted in the latest version of Part I of our papers (ll. 223-225 and 387-388) and discussed again in the Conclusion of Part II (ll. 344-355). See also point 3 of our responses above to Referee 2.

**4 Reference:**

Lorenz, E. N.:, Predictability: A problem partly solved. In: Proc. Seminar on Predictability, Vol. 1. ECMWF: Reading, Berkshire, UK, pp. 1–18, 1996.